# A new view on the origin of zero-bias anomalies of Co atoms atop noble metal surfaces

Juba Bouaziz [1✉], Filipe Souza Mendes Guimarães [1] & Samir Lounis [1✉]

Many-body phenomena are paramount in physics. In condensed matter, their hallmark is considerable on a wide range of material characteristics spanning electronic, magnetic, thermodynamic and transport properties. They potentially imprint non-trivial signatures in spectroscopic measurements, such as those assigned to Kondo, excitonic and polaronic features, whose emergence depends on the involved degrees of freedom. Here, we address systematically zero-bias anomalies detected by scanning tunneling spectroscopy on Co atoms deposited on Cu, Ag and Au(111) substrates, which remarkably are almost identical to those obtained from first-principles. These features originate from gaped spin-excitations induced by a finite magnetic anisotropy energy, in contrast to the usual widespread interpretation relating them to Kondo resonances. Resting on relativistic time-dependent density functional and many-body perturbation theories, we furthermore unveil a new many-body feature, the spinaron, resulting from the interaction of electrons and spin-excitations localizing electronic states in a well defined energy.

[1] Peter Grünberg Institut and Institute for Advanced Simulation, Forschungszentrum Jülich and JARA, Jülich 52425, Germany. ✉email: j.bouaziz@fz-juelich.de; s.lounis@fz-juelich.de

Signatures of many-body phenomena in solid state physics are diverse[1–5]. One of them is the Kondo effect emerging from the interaction between the sea of electrons in a metal and the magnetic moment of an atom[6,7], whose signature is expected below a characteristic Kondo temperature $T_K$. One of its manifestations is a resistivity minimum followed by a strong increase upon reducing the temperature, as initially observed in metals doped with a low concentration of magnetic impurities[8]. When the latter are deposited on surfaces, they can develop Kondo resonances evinced by zero-bias anomalies, with various Fano-shapes[9,10] that are detectable by scanning tunneling spectroscopy (STS), as shown schematically in Fig. 1a. The discovery of such low-energy spectroscopic features by pioneering STS measurements[5,11–13] opened an active research field striving to address and learn about many-body physics at the sub-nanoscale. A seminal example is Co adatoms deposited on Cu, Ag and Au (111) surfaces[5,11,13–18], which develop a dip in the transport spectra, with a minimum located at a positive bias voltage surrounded by steps from either sides (Fig. 1b). Although being commonly called Kondo resonances, the hallmarks of the Kondo effect have so far not been established for those particular Co atoms, i.e. the disappearance of the Kondo resonance at temperatures above $T_K$ and the splitting of the feature after applying a magnetic field[7,12,19–21]. A huge progress was made in advanced simulations combining quantum impurity solvers or even GW with density functional theory (DFT) addressing Kondo phenomena for various impurities (see e.g. refs. [22–26]), often neglecting spin-orbit interaction. The electronic structure spectra of realistic systems do not reproduce, in general, the experimental ones.

In the current work, we provide an alternative interpretation for the observed zero-bias anomalies in Co adatoms deposited on Cu, Ag and Au(111) surfaces, utilizing a recently developed framework resting on relativistic time-dependent DFT (TD-DFT) in conjunction with many-body perturbation theory (MBPT). Similar results were found for Co adatoms on Cu, Ag(001) surfaces and Ti adatom on Ag(001), which are shown in Supplementary Figure 1 for the sake of brevity. Our first-principles simulations indicate that the observed features find their origin in inelastic spin-excitations (SE), as known for other systems[12,27–35], which are gaped SE owing to the magnetic anisotropy energy that favors the out-of-plane orientation of the Co moment. Therefore, the physics is dictated by relativistic effects introduced by the spin-orbit interaction. As illustrated in Fig. 1b, the resulting

theoretical transport spectra are nearly identical to the experimental ones, advocating for a non-Kondo origin of the features. This effect induces two steps, asymmetric in their height, originating from intrinsic spin-excitations, and leads to the typically observed shape in the differential conductance, thanks to the emergence in one side of the bias voltage of a new type of many-body feature: a bound state that we name spinaron, emanating from the interaction of the spin-excitation and electrons. Finally, we propose possible experiments that enable the verification of the origin of the investigated zero-bias anomalies.

## Results

We compare our theoretical data to measurements obtained with low-temperature STS and proceed with a three-pronged approach for the first-principles simulations. We start from regular DFT calculations based on the full-electron Korringa-Kohn-Rostoker (KKR) Green function[36,37] method, which is ideal to treat Co adatoms on metallic substrates. We continue by building the tensor of relativistic dynamical magnetic susceptibilities for the adatom-substrate complex, $\chi(\omega)$, encoding the spectrum of SEs[29,38,39]. Finally, the many-body self-energy, $\Sigma(\varepsilon)$, is computed accounting for the SE-electron interaction including the spin-orbit coupling. The Tersoff-Hammann approach[40] allows the access to the differential conductance via the ground-state LDOS decaying from the substrate to the vacuum, where the STS-tip is located, here assumed to be located at 6.3 Å above the adatom for the Cu(111) surface and 7.1 Å for the Ag and Au(111) ones. This is then used to evaluate the renormalization of the differential conductance because of the SEs. More details are given in the Methods section and Supplementary Note 1.

**Zero-bias anomaly of Co adatom on Au(111).** We discuss here the different ingredients leading to the spectrum shown in Fig. 1b, which was found to be in a remarkable agreement with the data of Ref. [17], in particular. The adatom on Au(111) surface carries a spin moment of 2.22 $\mu_B$ and a relatively large orbital moment of 0.43 $\mu_B$. The easy axis of the Co magnetic moment is out-of-plane favored by a substantial magnetic anisotropy energy (MAE) of 4.46 meV (Table 1). This opens a gap in the SE spectrum, as illustrated in Fig. 2a, which shows the density of transversal SEs describing spin-flip processes, $-\frac{1}{\pi}\,\mathrm{Im}\,\chi^{+-}(\omega)$. The SE arises at 6.8 meV, which is shifted from the expected ideal location, 8 meV, as obtained from $4\frac{\mathrm{MAE}}{M_{\mathrm{spin}}}$ because of dynamical corrections[39]. As a

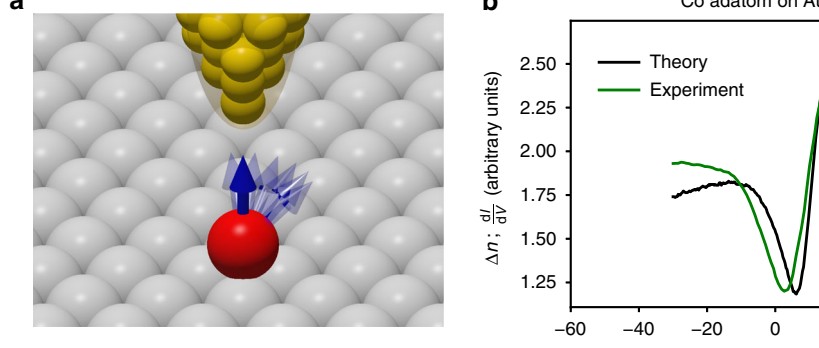

**Fig. 1 Scanning tunneling spectroscopy probing the differential conductance of a Co adatom. a** Illustration of the tip of a microscope scanning the surface of Au(111) on which an fcc Co adatom is deposited. **b** The differential conductance, d$I$/d$V$, measured at $T = 6$ K[17] compared to first-principles results. As deduced from the change of the LDOS owing to the presence of spin-excitations, $\Delta n$, the zero-bias anomaly stems from gaped spin-excitations and the presence of a many-body bound state, a spinaron, at positive bias voltage. The experimental data adapted with permission from IOP Publishing—Japan Society of Applied Physics—Copyright (2005).

result of electron-hole excitations of opposite spins[41,42], the lifetime $\tau$ of the SE is reduced down to 0.29 ps ($\tau = \frac{\hbar}{\Gamma}$, $\Gamma$ being the resonance width at half maximum). A simplified theory indicates that this effective damping is enhanced by the finite LDOS at the Fermi energy, which settles the density of electron-hole excitations[38]. The interaction of electrons and spin-excitations is incorporated in the so-called self-energy. It is represented by a complex quantity, with the real part shifting the energy of the electrons, and the imaginary part describing their inverse lifetimes. The significant components of the self-energy are spin diagonal, considering that the contribution of the off-diagonal elements is negligible for the investigated $C_{3v}$-symmetric adatom-substrate systems (see Supplementary Note 1). These quantities are computed from the dynamical susceptibilities $\chi(\omega)$ and the ground-state density $n_0(\varepsilon)$. For instance, the imaginary part for a given spin channel $\sigma$ after taking the trace over angular momentum indices, Im $\Sigma^{\sigma\sigma}(\varepsilon_F + V)$, is proportional to $\int_0^{-V} d\omega\, n_0^{\bar{\sigma}}(\varepsilon_F + V + \omega)$ Im $\chi^{\sigma\bar{\sigma},\bar{\sigma}\sigma}(\omega)$, i.e. it is a convolution of

the ground-state density, $n_0(\varepsilon)$, of the opposite spin-character and the SE density integrated over the bias voltage of interest. This quantity is plotted in Fig. 2b. Two steps are present, one for each spin channel, located at positive (negative) bias voltage for the minority (majority) self-energy. This is expected from the integration of the SE density over a resonance. As the self-energy of a given spin-channel is proportional to the LDOS of the opposite spin-character, the majority-spin self-energy has a higher intensity than the minority one, as expected from adatom LDOS illustrated in Fig. 2c, which decreases substantially the lifetime of the majority-spin electrons compared to that of minority-spin type.

The ground-state LDOS, $n_0^\sigma(\varepsilon)$, of Co adatom (depicted in Fig. 2c as black lines) varies very weakly for a bias voltage range of ~60 meV at the vicinity of the Fermi energy. The LDOS is then renormalized by the SE-electron interaction upon solving the Dyson equation

$$\underline{G}_R(\varepsilon) = [1 - \underline{G}(\varepsilon)\underline{\Sigma}(\varepsilon)]^{-1}\underline{G}(\varepsilon), \qquad (1)$$

from which the renormalized LDOS is obtained by tracing over site, spin and angular momenta of the Green function: $n(\varepsilon) = -\frac{1}{\pi}$ Im Tr $\underline{G}_R(\varepsilon)$.

At the adatom site (Fig. 2c), step-like features arise in the LDOS at the SE energy. The minority-spin LDOS hosts one single feature above the Fermi energy as expected from the corresponding self-energy. In contrast, the majority-spin LDOS is marked with an additional feature at positive voltage, which we identify as a many-body bound state — a spinaron. One can recognize it (spinaron) either from a one-to-one comparison between the spin-resolved LDOS and the self-energy, as being a feature not present in the latter one (see Supplementary Figs. 3 and 4) or

**Table 1 Basic calculated properties of Co adatom on Cu, Ag, and Au(111) surfaces.**

| Surface | $\tau$ (ps) | MAE (meV) | $M_{spin}(\mu_B)$ | $M_{orb}(\mu_B)$ |
|---|---|---|---|---|
| Cu(111) | 0.098 | 4.29 | 2.02 | 0.47 |
| Ag(111) | 0.088 | 3.27 | 2.21 | 0.70 |
| Au(111) | 0.073 | 4.46 | 2.22 | 0.44 |

Lifetime of the transverse spin excitations $\tau$, magneto-crystalline anisotropy (MAE), amplitude of the spin $M_{spin}(\mu_B)$ and orbital moments $M_{orb}(\mu_B)$. The positive sign of the MAE indicates that the magnetic moment points perpendicular to the three investigated substrates.

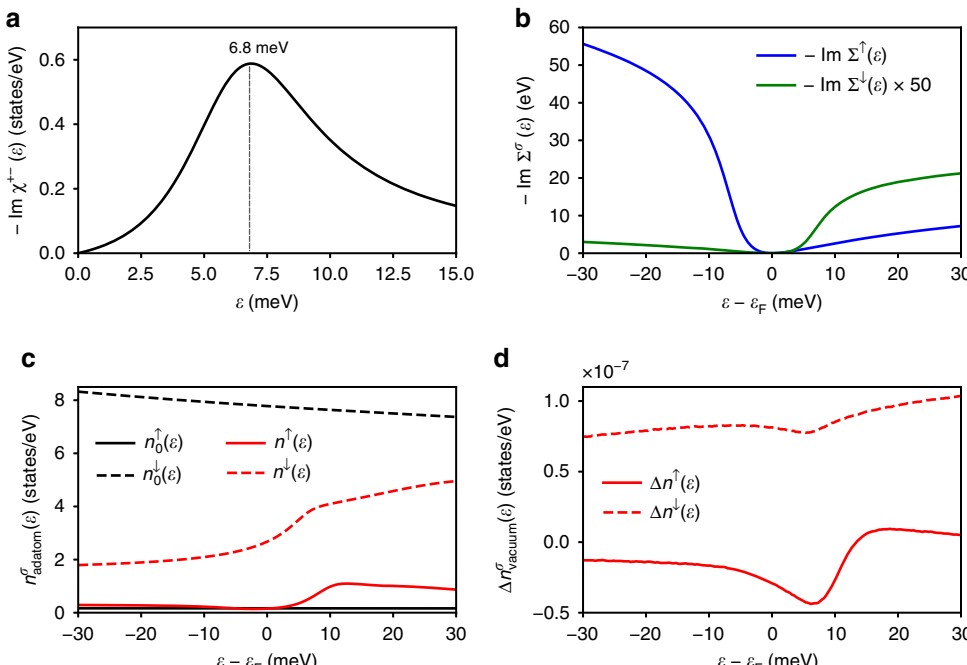

**Fig. 2 Spin-excitations, self-energies and local density of states for Co adatom/Au(111). a** Density of spin-excitations showing a broad resonance located at 6.8 meV, due to the adatom's magnetic anisotropy energy. The lifetime of the spin-excitation is 0.29 ps, which is mainly settled by electron-hole excitations. **b** Spin-resolved electronic self-energy $\Sigma$ after taking the trace over angular momentum indices, which inherits the information on the presence of the spin-excitation by hosting asymmetric steps above and below the Fermi energy. ↑ (↓) stands for majority-spin (minority-spin) . **c** LDOS at the adatom site before ($n_0$) and after ($n$) accounting for the interaction between electrons and the spin-excitation. **d** Spin-resolved change in the LDOS ($\Delta n$) calculated in the vacuum above the adatom. The minority-spin channel shows a low-intensity feature above the Fermi energy as expected from the corresponding self-energy. The large feature present in the majority-spin channel is composed of two steps, one originating from the intrinsic spin-excitation while the other is the spinaron, arising from the interaction of electrons and spin-excitations.

from tracking the intersections of Green functions and self-energies leading to the vanishing of the denominator of Eq. (1). The presence of spin-fluctuations affect the electronic behavior in terms of the electron-SE interaction encoded in the self-energy. This additional interaction can act as an attractive potential permitting the localization of electrons in a finite energy window, giving rise to a bound state. The spinaron emerges then when the denominator of the Dyson equation, Eq. (1), cancels out, i.e. when $\mathrm{Re}(\underline{G\Sigma}) = 1$, which occurs for the $d_{z^2}$ orbital having the ideal symmetry to be detected by STS, as illustrated in Supplementary Figs. 5 and 6. The spinaron is characterized by an energy and a lifetime (settled by $\mathrm{Im}(\underline{G\Sigma})$), both affected by the spin-orbit interaction, since it dictates the magnitude of the SE-gap defining the self-energy, and the electron-hole excitations.

The adatom electronic features decay into the vacuum, which are probed by the STS-tip in terms of the differential conductance. The signature of the SE is better seen in the change of the vacuum LDOS, $\Delta n = n - n_0$, illustrated in Fig. 1b and being spin-decomposed in Fig. 2d. One sees that the origin of the two steps and their asymmetry observed experimentally and theoretically is the concomitant contribution of the spin-excitation features and the spinaron. The signal is mainly emanating from the majority-spin LDOS, with the spinaron showing up as a step being higher than the one corresponding to the intrinsic SE below the Fermi energy. We note that the spinaron bears similarities to the spin polaron suggested to exist in halfmetallic ferromagnets[43].

**Systematic study of Co adatoms on Cu, Ag, Au(111) surfaces.** We performed a systematic comparison between simulated and experimental data and evidenced that the the spin-excitations combined with the spinaron are generic features for Co adatoms deposited on Cu(111), Ag(111) and Au(111). The agreement shown in Fig. 3a is staggering, certifying that the result obtained for Au(111) surface is not accidental. Similarly to Au, the spinaron originates from the $d_{z^2}$ on Ag and Cu, conferring the right symmetry to be detected efficiently with STS (more details are provided in Supplementary Figs. 5 and 6). This enforces our view that the experimentally observed zero-bias features for Co adatoms are captured by gaped SEs. The energies and lifetimes ($\varepsilon_{\mathrm{spinaron}}$, $\tau_{\mathrm{spinaron}}$) of the spinarons as obtained from the theoretical spectra in vacuum are: (4.42 meV, 0.34 ps), (12.6 meV, 0.20 ps) and (9.41 meV, 0.20 ps) for Cu, Ag, and Au(111), respectively, which are of the same order of magnitude than those of the intrinsic spin-excitations listed in Table 1. Interestingly, the lifetimes of the latter excitations increase slightly on Cu and Ag surfaces when compared to that obtained on Au. Interestingly, the spectra obtained for Co/Cu(111) are in line with those reported in ref. [48] based on a simplified theoretical approach (more details are provided in Supplementary Note 2).

Merino and Gunnarsson[49] suggested that the surface states of the investigated substrates give rise to the particular shape of the low-energy excitations. In the case of Ag(111), STS experiments showed the possible alteration of the tunneling signal depending on the scattering of the Ag surface state at surrounding defects and step edges[50]. When compared to other surfaces (see Supplementary Fig. 1), our theory indicates that the surface states are important to enhance the overall signal in the vacuum while the main origin of the spectral anomalies of the isolated adatoms is a combination of the intrinsic spin-excitations

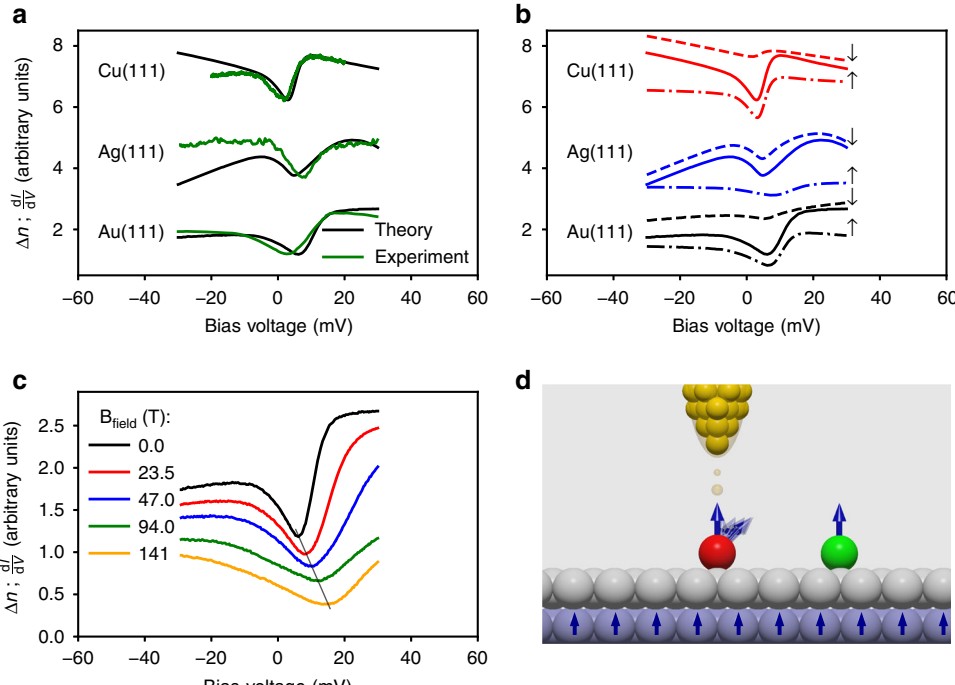

**Fig. 3 Systematic tunneling transport spectra of Co adatoms on Cu, Ag and Au(111) surfaces. a** Excellent agreement between the first-principles spectra and those measured with STS at temperatures of 1.2 K[13], 1.1 K[44], 6 K[17]. **b** Spin-resolved spectra indicating that spin-polarized spectroscopy can reshape the measured differential conductance, which would help to disentangle the various contributions to the spectra. The total (solid) signal is obtained by the sum of both spin channels, which were shifted for a better visualization and comparison. **c** Response of the zero-bias anomalies to magnetic fields. The spin-excitation gap opens while the dip moves to larger energies. Large magnetic field are possible in some STS setups (14 T[45] or even 38 T[46,47]). **d** Proximity effects can be used via neighboring adatoms and by depositing thin films of noble metals on a ferromagnetic substrate, where the magnetic exchange interaction felt by the adatom acts as an effective magnetic field. The experimental data in **a** adapted with permission from IOP Publishing for Cu[13] and from the Japan Society of Applied Physics—Copyright (2005) for Au[17] as well as from Nature Publishing Group for Ag[44].

signatures and the spinaron. The weight and shape of each of the features depend on the substrate and interference effects induced by decay of the electronic states of both the adatom and surface. Moreover Eq. (1), shows that both the step-like and peak-like features of the respective imaginary and real parts of the self-energy are mixed up, contributing to the signature of the observed low-energy anomalies (see Supplementary Fig. 3). One sees in Fig. 3b, that the main difference between the three surfaces originates from intrinsic spin-excitation occurring in the minority-spin channel (positive bias voltage), which is for Ag more enhanced than the signal coming from the majority-spin channel carrying both the intrinsic spin-excitation (negative bias voltage) and the spinaron (positive bias voltage). When deposited on Cu and Au, the asymmetry between majority- and minority-spin channels switches. This is induced by the electronic structure of the adatom on the three surfaces (see Supplementary Fig. 2). The LDOS at the Fermi energy of Co on Ag hosts a larger minority-spin DOS than on Cu and Au. The reason is the weaker hybridization strength between the electronic states of Co and the substrate, when compared to Cu or Au(111), which reduces the broadening of the minority-spin resonance on the former. The positions of the steps pertaining to the intrinsic SEs correlate with the magnitude of the MAE, which favors the out-of-plane orientation of the Co magnetic moment on the three substrates as listed in Table 1.

For the quantitative validation of the agreement between the theoretical and experimental data, we fit our data with the commonly used Fano-resonance formula for the differential conductance of Kondo resonances[9,10,15]:

$$n(\varepsilon) = \mathcal{A}\,\frac{(\varepsilon + q)^2}{\varepsilon^2 + 1}\quad, \qquad (2)$$

with $\mathcal{A}$ being the amplitude of the signal and $q$ the coupling parameter. The latter plays an important role in the Fano formalism as it determines the shape and asymmetry of the STS-signal. $\varepsilon = (eV - E_0)/k_B T_K^{\mathrm{eff}}$ encodes the information regarding the effective Kondo temperature $T_K^{\mathrm{eff}}$, as well as the bias voltage $V$ (with $k_B$ being the Boltzmann constant and $E_0$ the position of the investigated resonance). The fitted Fano-parameters are listed in Table 2. Astonishingly, the recovered effective temperatures are in perfect agreement with the ones obtained from experimental data.

**Experimental proposals: impact of spin-polarized tip, magnetic field, and proximity-effects**. As mentioned before, the Kondo origin of the low-energy spectral features has so far not been evidenced for the systems investigated in the current work. This is usually realized by performing temperature-dependent measurements and/or upon the application of a magnetic field, which would respectively result in a broadening of the anomalies and/or their splitting. However, the large broadening of the dip-like structure require improved energy resolutions than currently available, preventing the realization of such experimental studies.

Here we address possible experiments that can further verify our predictions.

Kondo resonances should not change when probed by a spin-polarized tip. Our spectra are however spin-dependent and thus we expect the alteration of their shape depending on the spin-polarization of the tip, $P_{\mathrm{tip}} = \frac{n_{\mathrm{tip}}^{\uparrow} - n_{\mathrm{tip}}^{\downarrow}}{n_{\mathrm{tip}}^{\uparrow} + n_{\mathrm{tip}}^{\downarrow}}$, since the differential conductance is approximately proportional to $(1 + P_{\mathrm{tip}})n_{\mathrm{adatom}}^{\uparrow} + (1 - P_{\mathrm{tip}})n_{\mathrm{adatom}}^{\downarrow}$[40,51]. Ultimately, manipulating the spin-polarization of the tip (see e.g. Ref. [52]) would help spin-resolving the LDOS as depicted in Fig. 3b for the three investigated substrates.

Furthermore, the zero-bias dip is expected to split into two features for a traditional Kondo resonance once an external magnetic field is applied[7]. Figure 3c shows a completely different behavior. The field applied along the easy axis of the Co atoms yields an increase of the excitation gap, as expected, and of the spinaron energy (see the spin-resolved spectra in Fig. S1). The interplay of the various features gives the impression that the observed dip drifts to energetically higher unoccupied states, which occurs because of the presence of the spinaron. We note that applying a magnetic field in the direction perpendicular to the magnetic moment, would affect the excitation gap in a non-trivial way[32]. A field of 14 Tesla is available in some STS setups[45] and can even reach 38 Tesla[46,47]. Larger fields can be accessed effectively via magnetic-exchange-mediated proximity effect by either (i) bringing another magnetic atom to the vicinity of the probed adatom or (ii) depositing the probed adatom on a magnetic surface with a non-magnetic spacer in-between (see Fig. 3d). If the adjacent atom is non-magnetic, it can modify the MAE, which dictates the magnitude of the SE gap. If the MAE is reduced, the lifetime of the spin-excitations is expected to increase, since the amount of electron-hole excitations available in the respective energy range would decrease. This can then favor the monitoring of the impact of temperature and magnetic field on the zero-bias anomalies, helping to distinguish a Kondo behavior from the one emerging from spin-excitations.

## Discussion

The zero-bias anomalies probed by low-temperature scanning tunneling spectroscopy on Co atoms deposited on Cu, Ag, and Au (111) surfaces, usually identified as Kondo resonances, are shown to be the hallmarks of gaped spin-excitations enhanced by the presence of spinarons. We note that there are other examples of materials, such as quantum wires, where zero-bias anomalies have been challenged to be Kondo features[53,54]. However, the adatoms investigated in the current work represent the most traditional systems, where the surface science community converges to the Kondo-related interpretation. The gap of the spin-excitations is induced by the magnetic anisotropy energy of the Co adatom, defining the meV energy scale requested to excite the magnetic moment, and therefore its magnitude can be extracted from the position of the observed steps. Considering that the large

**Table 2 Fitted Fano parameters for a single Co adatom deposited on Cu, Ag, and Au(111) surfaces.**

| Surface | $E_0$(meV) | $T_K^{\mathrm{eff}}$(K) | $T_K^{\mathrm{exp}}$(K) | $q$ | $q^{\mathrm{exp}}$ |
|---|---|---|---|---|---|
| Cu(111) | 3.74 | 37.3 | 44.9 [57[17]] | 0.42 | 0.38 [0.5[13]] |
| Ag(111) | 4.71 | 89.4 | 73 [56[44,50]] | −0.05 | −0.004 [0.02 ± 0.02[44,50]] |
| Au(111) | 10.61 | 67.5 | 91 [76 ± 8[17]] | 0.55 | 0.45 [0.7[17]] |

The resonance formula used for fitting is given in Eq. (2). The effective Kondo temperature $T_K^{\mathrm{eff}}$ and the coupling parameter $q$ extracted from our own fits of the experimental and theoretical spectra are compared to those published in refs. [13,17,44,50] (values between brackets).

magnetic moments of the Co adatoms are characterized by an out-of-plane easy axis, Kondo-screening is unlikely to occur[21], and enforces the view that the zero-bias anomalies result from spin-excitations. Additional simulations performed on Co adatoms on Cu and Ag(001) surfaces as well as Ti adatom on Ag (001), shown in Supplementary Fig. 1, provide additional evidence that spin-excitations are potentially present on other materials, giving rise to the experimentally observed zero-bias anomalies.

Grounding on a powerful theoretical framework based on relativistic time-dependent density functional and many-body perturbation theories, we obtain differential conductance spectra reproducing extremely well the measured data. We systematically demonstrate the presence of spinarons, which are many-body bound-states emerging from the interaction of electrons and spin-excitations. While the self-energies quantifying the interaction of the electrons and spin-excitations are dynamical in nature and account for various correlation effects, it would be interesting to prospect in the future the impact of correlations (in the spirit of DFT + U[55]) on the ground state properties, such as the magnetic anisotropy energy and subsequently on the excitation behavior of the investigated materials. In general, our findings call for a profound change of our understanding of measured zero-bias anomalies of various nanostructures, which stimulates further theoretical developments permitting the ab-initio investigation of Kondo features, spinarons, spin-excitations, and spin-orbit driven physics on equal footing.

The one-to-one agreement between our first-principles spectra and the available experimental ones strongly advocates for the importance of the spin-excitations in the interpretation of the origin of the zero-bias anomalies. X-ray magnetic circular dichroism (XMCD) experiments should help to confort our findings by unveiling the magnetic nature as well as the magnetic anisotropy energy of the investigated adatoms as done for Co adatoms on Pt(111)[56]. Surprisingly, this was, so far, not performed. Temperature-dependent and magnetic-field STM-based measurements were, to our knowledge, not reported, which is explained by the extreme difficulty to probe with enough resolution modifications induced in the rather broad spectral features. We conjecture that this might change in the near future, for example with electron-spin-resonance STM (ESR-STM)[57,58] if realized on metallic substrates. In this work, various experimental setups were proposed, which would permit to further confirm our predictions. For instance, the theoretical spectra are spin-dependent and therefore the weight of each spin-channel to the total STM spectrum should depend on the spin-polarization of the tip. Furthermore, the application of a magnetic field is expected to increase the gap of the intrinsic spin-excitations, while a splitting is expected for Kondo features. However, the presence of the spinaron leads to an unconventional behavior, that is the excitation gap increases but the effective dip is not fixed and shifts to larger bias voltages. Currently, a few STM setup allow to reach large magnetic fields (e.g., 14 T and even 38 T), which would be enough to check our predictions. But even if those fields are not available, a reasonable alternative would be to use the proximity-induced effective magnetic field emerging from an adjacent magnetic adatom. Finally, one could tune down the magnetic anisotropy energy in order to reduce the amount of electron-hole excitations that are responsible for the broadening of the spin-excitations. This could be realized by attaching a non-magnetic atom such as Cu, for example, to Co adatom, after which the experimental investigation of the impact of temperature and magnetic fields would become more amenable.

By opening a new perspective on low-energy spectroscopic features characterizing subnanoscale structures deposited on substrates, built upon the pioneering work of the STS community (see e.g. refs. [5,11–13]), our findings motivate new experiments exploring the interplay of temperature, proximity effects, and response to an external magnetic field, which can help identifying the real nature of the observed excitations and unravel the complexity and richness of the physics behind the spinaron.

## Methods
Our first-principles approach is implemented in the framework of the scalar-relativistic full-electron Korringa-Kohn-Rostoker (KKR) Green function augmented self-consistently with spin-orbit interaction[36,37], where spin-excitations are described in a formalism based on time-dependent density functional theory (TD-DFT)[39,41,59,60] including spin-orbit interaction. Many-body effects triggered by the presence of spin-excitations are approached via many-body perturbation theory[48,61,62] extended to account for relativistic effects. The method is based on multiple-scattering theory allowing an embedding scheme, which is versatile for the treatment of nanostructures in real space. The full charge density is computed within the atomic-sphere approximation (ASA) and local spin density approximation (LSDA) is employed for the evaluation of the exchange-correlation potential[63]. We assume an angular momentum cutoff at $l_{max} = 3$ for the orbital expansion of the Green function and when extracting the local density of states a k-mesh of $300 \times 300$ is considered. The Co adatoms sit on the fcc stacking site relaxed towards the surface by 20% (14%) of the lattice parameter of the underlying Au and Ag (Cu) substrates.

After obtaining the ground-state electronic structure properties, the single-particle Green functions are then employed for the construction of the tensor of dynamical magnetic susceptibilities, $\chi(\omega)$, within time-dependent density functional theory (TD-DFT)[29,38,39] including spin-orbit interaction. The susceptibility is obtained from a Dyson-like equation, which renormalizes the bare Kohn-Sham susceptibility, $\underline{\chi}_{KS}(\omega)$ as

$$\underline{\chi}(\omega) = \underline{\chi}_{KS}(\omega) + \underline{\chi}_{KS}(\omega)\,\underline{\mathcal{K}}\,\underline{\chi}(\omega). \quad (3)$$

$\underline{\chi}_{KS}(\omega)$ describes uncorrelated electron-hole excitations, while $\underline{\mathcal{K}}$ represents the exchange-correlation kernel, taken in adiabatic LSDA (such that this quantity is local in space and frequency-independent[64]). The energy gap in the spin excitation spectrum is accurately evaluated using a magnetization sum rule[29,38,39].

## Data availability
All data needed to evaluate the conclusions in the paper are present in the paper and/or the Supplementary Information. Additional data related to this paper may be requested from the authors.

## Code availability
The KKR Green function code that supports the findings of this study is available from the corresponding author on reasonable request.

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

## Acknowledgements
We thank Markus Ternes, Alexander Weismann, Nicolas Lorente and Wolf-Dieter Schneider for fruitful discussions. We are grateful to Michael Crommie, Lars Diekhöner, Peter Wahl, Alexander Schneider, Markus Ternes, Klaus Kern for sharing with us their original data measured with scanning tunneling microscopy. This work is supported by the European Research Council (ERC) under the European Union's Horizon 2020 research and innovation programme (ERC-consolidator grant 681405 - DYNASORE). We acknowledge the computing time granted by the JARA-HPC Vergabegremium and VSR commission on the supercomputer JURECA at Forschungszentrum Jülich.

## Author contributions
S.L. initiated, designed and supervised the project. J.B. developed the theoretical ab-initio scheme accounting for spin-orbit interaction in the calculation of the many-body self-energies. J.B. performed the simulations and F.S.M.G. contributed to data post-processing. All authors discussed the results and helped writing the paper.

## Funding

## Competing interests
The authors declare no competing interests.
