## [Peer Review File · Nature Communications]

REVIEWER COMMENTS

Reviewer #1 (Remarks to the Author):

Report on “A new view on the origin of zero-bias anomalies of Co atoms atop noble metal surfaces” by Bouaziz et al.

This manuscript is an interesting well-written new interpretation of the Co zero-bias anomaly in tunnel spectra. The authors have developed a new tool to compute low-energy excitations from ab-initio electronic-structure calculations and this allows them to look into the very interesting physics of zero-bias anomalies from a different perspective. As a consequence, the authors find new spin excitations that could account for the zero-bias anomaly that have never been considered before. The authors convincingly argue that the spectra are due to this new type of localized excitation produced in an electron gas by a local spin. From my point of view, this work is an important step forward in the understanding of spin excitations on metals and on the type of information that the advanced STM spectra can access. I think this work should be published as soon as possible.

The theoretical treatment does not seem to include the type of correlations needed for the Kondo physics that until now has been used to explain the zero-bias anomaly. However, Kondo in the weak-coupling regime can be easily treated by perturbative approaches or approaches lacking correlations such as the GW approach. From this point of view, I wonder if the authors are not obtaining a weak Kondo signal instead of a localized spin excitation. The work showing the Kondo anomaly using GW could be interesting to cite and compare to: PRB 77, 115333 (2008) By Thygesen and Rubio.

The work by Merino and Gunnarsson (PRL 93, 156601 (2004)) and more recently Ref. 45 of the manuscript, convincingly argue about the role of the surface state as causing the dip that rather looks like a spin excitation. I have missed a discussion of the Merino work, and the role of the surface state in the present spectra.

In summary, the authors are bringing in fresh air into an old problem that is still an open issue. I do recommend publication of the manuscript after the authors have considered my above optional remarks.

Reviewer #2 (Remarks to the Author):

Dear Editor

The manuscript by Bouaziz and co-workers "A new view on the origin of zero-bias anomalies of Co atoms atop noble metal surfaces" reports on theoretical study of spin excitations (SE) of Co ad-atoms deposited on

Au(111), Cu(111), and Ag(111) surfaces. A special emphasis is made on their interactions with itinerant

conduction electrons of the substrate. It is shown, in particular, that the spectral function (or the density of states, DOS)

of conduction electrons gets strongly renormalized around the Fermi energy due to their interaction with Cobalt SE.

The calculated further local DOS in vacuum, which can be directly probed by an STM tip, reproduces rather

well zero-bias anomalies (ZBA) in differential conductance observed generally (as dips) for Co ad-atoms on (111) surfaces.

The authors therefore argue that experimental ZBA are manifestations of inelastic electron scattering on low energy Cobalt SE rather than

of the Kondo physics, a scattering on a many-body Kondo resonance forming right at the Fermi energy due to correlations on Co orbitals.

Though I find results rather interesting I can not unfortunately recommend the manuscript for publication, at least in the

present form. I present below few points under question and some remarks.

1) Novelty: the manuscript follows previous authors's papers, mainly Ref. 37 (PRB 91, 075405) and Ref. 48 (PRB 89, 235439),

extending the calculation of electron self-energies of Ref. 48 to account explicitly for spin-orbit effects which were simulated

in Ref. 48 by an auxiliary magnetic field.

On my opinion not much new physics is contained in the present work compared to results of Ref. 48 --

very similar shapes of magnetic response functions, electron self-energies, and DOS were already reported

in Figs. 2, 3 and 5 of Ref. 48 for a Co on Cu(111), together with some other cases, the effect of magnetic field was also investigated.

I presume that the collective "spinaron" state (introduced in the present manuscript)

was also present in Ref. 48 though not discussed explicitly.

The authors, on my opinion, should state therefore more clearly if inclusion of spin-orbit effects is crucial and brings any new physics. Or it just

corrects for more accurate magnetic anisotropy energy (MAE), related SE energies, etc. with respect to Ref. 48. Also if spin-orbit is accounted for

by additional term in the Hamiltonian, like in Eq. 3 of Ref 37, it is controlled anyway by an additional parameter ξ which should be also fixed somehow.

The choice of this parameter should be also mentioned and discussed.

2) Novelty and reliability: As far as I know the nature of ZBA in conductance has a rather long discussion in the surface science community, there are many cases

where these features, ascribed initially to Kondo-related physics, were then attributed to some inelastic processes (mainly to a scattering

on low energy phonons but also on magnetic excitations), so it is still under discussion. On the other side, the Co ad-atoms represent the most traditional

case where the community seems to converge to the Kondo-related interpretation.

I think therefore that the authors should be more modest and present their results as another indication towards non-Kondo interpretation, not denying

however completely the Kondo-related realization. The two effects, Kondo screening and inelastic electron/spin scattering, could possibly even co-exist..

If the authors really want to discard the Kondo interpretation, they should provide some more arguments. For example,

the spin $S > 1/2$ of Cobalt (it is $S=1$) together with finite MAE could prevent for the Kondo screening of the Cobalt spin.

In this respect, I would strongly suggest to look at the Cu(001) surface (and other (001) surfaces)

where very different (from presented here (111) orientations) line-shapes were reported experimentally (Ref. 11).

If Kondo fluctuations for Co are frozen by above arguments, the presented here "spin excitation" hypothesis should also explain and reproduce

experimental asymmetric line-shapes.

3) I find the discussion on "spinaron" rather superficial, could the authors provide a more detailed description of what it is and

how do they find it. Probably they could provide a figure with real and imaginary parts of self-energy? Also I would appreciate some more

details on its energies and life times for different substrates.

I wonder also if there is any relation between "spinaron" and Kondo resonance since they are both collective states made of

conduction electron+local spin. Can "spinaron" be considered as a kind of "approximate" Kondo state approached within mean-field DFT method?

Does the "spinaron" exist also if MAE is zero (gapless spin excitations) or it needs a finite anisotropy (gapped SE)?

4) Could the authors explain in more detail how do they construct vacuum DOS, at which distance above the surface, what is orbital composition

of it (s, p, d)? Why in Fig. 2d the authors present the changes of LDOS but not the total ones which I think should be probed by an STM tip?

Moreover, in Fig. 3b it seems that the solid lines are not the averages, they do not lie in between dashed lines (up and down).

5) I wonder also about the effect of possible "Hubbard" +U corrections. In static DFT calculation U usually moves significantly

d-states from the Fermi energy (occupied states down and empty up in energy). Since the self-energies are the convolutions of DFT d-states DOS

and response function, I would expect that d-DOS modification at the Fermi energy due to +U corrections would also affect importantly the results.

Reviewer #3 (Remarks to the Author):

This paper reports on tunneling spectra taken by scanning tunneling microscopy (STM) on single Co atoms adsorbed on noble metal (111) surfaces. Whereas those spectra have been so far interpreted as a Fano resonance of the Kondo resonance, the present paper claims they are explained with inelastic tunneling (spin-flipping) and the interaction of the excited spin with substrate electrons based on the authors' theoretical calculations.

The presented agreements with experimental results are good, but indeed it was surprising to me whether and why this kind of rather simple interpretation really works. If this is really true several questions arise as mentioned below, and I would like to judge whether the paper should be accepted or not in Nature Communications after the questions are clearly solved.

In this paper, the authors claim quite large magnetic anisotropic energy (MAE) in these systems. This is quite surprising to me because so far only Co/Pt(111) system is known to have large MAE. As authors mentioned in this manuscript, in order to have MAE spin-orbit interaction (SOI) plays a crucial role (2nd order perturbation). It is then strange to me why the three substrates in particular Cu and Au exhibit similar values of MAE. Au should have much larger SOI and thus MAE as a heavy element. The spin relaxation time by Fermi sea should also be dependent on SOI but three substrates give similar lifetime values. These seem strange to me.

The authors mention inelastic spin flip energy is given as $4(\text{MAE})/M_{\text{spin}}$. It is also strange to me why it is not simply MAE. Why is M_{spin} involved here?

It has been known that single Co atoms adsorbed on Pt(111) substrate show a large MAE. Spectra showing inelastic tunneling were reported, but, not Kondo-like features, which seems inconsistent with the model presented in this manuscript. What is the difference between the three substrates and Pt, and how the difference is induced?

Whether the model is correct or not can be easily checked experimentally, by measuring temperature dependence of the spectra, or performing spin-polarized tunneling spectroscopy on the adatoms. I wonder why authors do not collaborate with experimental groups. Since most of the Kondo resonance works were done at 4K so far, performing at lower temperature is enough to distinguish the two interpretations, I guess.

In the manuscript, the authors provide rather detailed explanations on generally known things, such as energy shift/broadening of self-energy, renormalization of Green's function etc. On the other hand, physical explanations on the calculated results, such as why the three substrates have different spectra, what are the spin states and orbital occupation of adsorbed Co atoms, why background slopes of the three spectra are different (Cu and Au go up with the voltage but Ag goes down) etc. are missing. These are quite important information for understanding the physics and interpretation of the model and should not be missed.

Prof. Dr. Samir Lounis

Chair of theoretical nanospintronics at RWTH-Aachen
Leader of Functional Nanoscale Structure Probe and
Simulation Laboratory

Tel.: 02461-61-4068

E-Mail: s.lounis@fz-juelich.de

www.fz-juelich.de/pgi/Group-Lounis

PGI-1/IAS-1: Quantum Theory of Materials

First Reviewer

Report on “A new view on the origin of zero-bias anomalies of Co atoms atop noble metal surfaces” by Bouaziz et al.

This manuscript is an interesting well-written new interpretation of the Co zero-bias anomaly in tunnel spectra. The authors have developed a new tool to compute low-energy excitations from ab-initio electronic-structure calculations and this allows them to look into the very interesting physics of zero-bias anomalies from a different perspective. As a consequence, the authors find new spin excitations that could account for the zero-bias anomaly that have never been considered before. The authors convincingly argue that the spectra are due to this new type of localized excitation produced in an electron gas by a local spin. From my point of view, this work is an important step forward in the understanding of spin excitations on metals and on the type of information that the advanced STM spectra can access. I think this work should be published as soon as possible.

Answer 1.1: We thank the reviewer for his/her positive assessment of our manuscript. We are glad that he/she finds our work to be an important step forward in this field and thank him/her for the strong enthusiasm in publishing it as soon as possible.

The theoretical treatment does not seem to include the type of correlations needed for the Kondo physics that until now has been used to explain the zero-bias anomaly. However, Kondo in the weak-coupling regime can be easily treated by perturbative approaches or approaches lacking correlations such as the GW approach. From this point of view, I wonder if the authors are not obtaining a weak Kondo signal instead of a localized spin excitation. The work showing the Kondo anomaly using GW could be interesting to cite and compare to: PRB 77, 115333 (2008) By Thygesen and Rubio.

Answer 1.2: We thank the Reviewer for pointing out the article of Thygesen and Rubio. It is reassuring to see that GW applied on the Anderson model, as done in PRB 77, 115333 (2008), can give rise to a zero-bias anomaly for a spin 1/2 and no spin-orbit coupling. The spectra resulting from our simulations originate primarily from the presence of spin-excitations. This leads to a pair of features at positive and negative bias voltage, whose position is controlled by the magnitude of the magnetic anisotropy energy, i.e. the effect originates from the relativistic spin-orbit interaction, which opens the excitation gap. The magnitude and shape of these two features is not necessarily identical in shape and intensity. The final ingredient that gives the particular step-like shape of the theoretical spectra is the spinaron, which results from the interaction of the spin-excitations and the electrons. The spinaron is not necessarily located at zero-bias, or close to it. In the case of Co on the three investigated surfaces, it emerges in the majority-spin channel, not in the minority-spin one. Therefore, we refrain from referring to the spinaron as a weak Kondo effect.

We modified the last part of the first paragraph of the main text adding a reference to the article of Thygesen and Rubio: “A huge progress was made in advanced simulations combining quantum impurity solvers or

Prof. Dr. Samir Lounis

Chair of theoretical nanospintronics at RWTH-Aachen
Leader of Functional Nanoscale Structure Probe and
Simulation Laboratory

Tel.: 02461-61-4068

E-Mail: s.lounis@fz-juelich.de

www.fz-juelich.de/pgi/Group-Lounis

PGI-1/IAS-1: Quantum Theory of Materials

even GW with density functional theory (DFT) addressing Kondo phenomena for various impurities (see e.g. Refs. [22-26]), often neglecting spin-orbit interaction. The electronic structure spectra of realistic systems do not reproduce, in general, the experimental ones.”

The work by Merino and Gunnarsson (PRL 93, 156601 (2004)) and more recently Ref. 45 of the manuscript, convincingly argue about the role of the surface state as causing the dip that rather looks like a spin excitation. I have missed a discussion of the Merino work, and the role of the surface state in the present spectra.

Answer 1.3: We noticed, when comparing to other surfaces, that the bare signal from the substrate is enhanced thanks to the presence of the surface state, which facilitates the theoretical observation of the inelastic spectra. The case of 001 surfaces of Cu and Ag, which do not host surface states is now shown in Supplement Figure 1 (Figure 2 of this reply) at the demand of Reviewer 2 (Answer 2.3). The main origin of the shapes of the zero-bias anomalies, as they come from our theory, is the combination of the intrinsic spin-excitations signatures and the spinaron. The weight of each of the features is different when comparing the spectra obtained on Cu(111) and Au(111) to those obtained for Ag(111), in excellent agreement with the experiment. One sees in Fig. 3b of the main text that this originates from intrinsic spin-excitation occurring in the minority-spin channel (positive bias voltage), which is stronger for Ag than the signal coming from the majority-spin channel carrying both the intrinsic spin-excitation (negative bias voltage) and the spinaron (positive bias voltage). When deposited on Cu and Au, the asymmetry between majority- and minority-spin channel switches. This is induced by the electronic structure of the adatom on the three surfaces (see Figure 1, now shown in the Supplement Figure 2). The minority-spin local density of states (LDOS) at the Fermi energy is larger for Co on Ag(111) than on Cu(111) and Au(111). The reason is the weaker hybridization strength between the electronic states of Co and the substrate, when compared to Cu or Au(111), which reduces the broadening of the minority-spin resonance on the former.

One of the goals of Ref. [M. Moro-Lagares et al, Phys. Rev. B 97, 235442 (2018)] (previously numbered [45], now [47]) is to explain the origin of the large range of effective Kondo temperatures extracted for Co adatoms on Ag(111). The reason invoked — and that’s the essence of the paper — is that this is induced by the presence of defects and edges that scatter back the surface state of Ag, which is close to the Fermi energy. Considering that minority-spin state of Co/Ag(111) is sharp and close to the Fermi energy as obtained from our calculations, the electronic and magnetic responses can be large and sensitive to perturbations like those considered in the experimental part of that paper. This is a problem worth investigating in the near future. Interestingly, it is mentioned in the paper that the temperature-dependent study of the resonances did not lead to an exponential dependence, as expected for Kondo resonances.

Following the advice of the Reviewer, we comment on both papers and address more clearly the origin of the shape of the ZBA as they come from our theory. Here are our amendments to the main text: “Merino and Gunnarsson [46] suggested that the surface states of the investigated substrates give rise to the particular shape of the low energy excitations. In the case of Ag(111), STS experiments showed the possible alteration of the tunneling signal depending on the scattering of the Ag surface state at surrounding defects and step

Prof. Dr. Samir Lounis

Chair of theoretical nanospintronics at RWTH-Aachen
Leader of Functional Nanoscale Structure Probe and
Simulation Laboratory
Tel.: 02461-61-4068
E-Mail: s.lounis@fz-juelich.de
www.fz-juelich.de/pgi/Group-Lounis

Figure 1: **Total spin-polarized local density of states of a Co adatom on Cu, Ag and Au(111) surfaces.** The minority-spin state for Co on Ag(111) is sharper due to the weaker strength of the hybridization of the electronic states of the adatom and the substrate, when compared to Au or Cu(111).

edges. When compared to other surfaces (see Supplementary Figure 1), our theory indicates that the surface states are important to enhance the overall signal in the vacuum while the main origin of the spectral anomalies of the isolated adatoms is a combination of the intrinsic spin-excitations signatures and the spinaron. The weight and shape of each of the features depend on the substrate and interference effects induced by decay of the electronic states of both the adatom and surface. Moreover Eq. 1, shows that both the step-like and peak-like features of the respective imaginary and real parts of the self-energy are mixed up, contributing to the signature of the observed low-energy anomalies (see Supplementary Figure 3). One sees in Figure 3b, that the main difference between the three surfaces originates from intrinsic spin-excitation occurring in the minority-spin channel (positive bias voltage), which is for Ag more enhanced than the signal coming from the majority-spin channel carrying both the intrinsic spin-excitation (negative bias voltage) and the spinaron (positive bias voltage). When deposited on Cu and Au, the asymmetry between majority- and minority-spin channels switches. This is induced by the electronic structure of the adatom on the three surfaces (see Supplementary Figure 2). The LDOS at the Fermi energy of Co on Ag hosts a larger minority-spin DOS than on Cu and Au. The reason is the weaker hybridization strength between the electronic states of Co and the substrate, when compared to Cu or Au(111), which reduces the broadening of the minority-spin resonance on the former. ”

In summary, the authors are bringing in fresh air into an old problem that is still an open issue. I do recommend publication of the manuscript after the authors have considered my above optional remarks.

Prof. Dr. Samir Lounis

Chair of theoretical nanospintronics at RWTH-Aachen
Leader of Functional Nanoscale Structure Probe and
Simulation Laboratory

Tel.: 02461-61-4068

E-Mail: s.lounis@fz-juelich.de

www.fz-juelich.de/pgi/Group-Lounis

PGI-1/IAS-1: Quantum Theory of Materials

Answer 1.4: We thank the Reviewer once again for his/her positive assessment.

Prof. Dr. Samir Lounis

Chair of theoretical nanospintronics at RWTH-Aachen
Leader of Functional Nanoscale Structure Probe and
Simulation Laboratory

Tel.: 02461-61-4068

E-Mail: s.lounis@fz-juelich.de

www.fz-juelich.de/pgi/Group-Lounis

PGI-1/IAS-1: Quantum Theory of Materials

Second Reviewer

Dear Editor The manuscript by Bouaziz and co-workers “A new view on the origin of zero-bias anomalies of Co atoms atop noble metal surfaces” reports on theoretical study of spin excitations (SE) of Co ad-atoms deposited on Au(111), Cu(111), and Ag(111) surfaces. A special emphasis is made on their interactions with itinerant conduction electrons of the substrate. It is shown, in particular, that the spectral function (or the density of states, DOS) of conduction electrons gets strongly renormalized around the Fermi energy due to their interaction with Cobalt SE. The calculated further local DOS in vacuum, which can be directly probed by an STM tip, reproduces rather well zero-bias anomalies (ZBA) in differential conductance observed generally (as dips) for Co ad-atoms on (111) surfaces. The authors therefore argue that experimental ZBA are manifestations of inelastic electron scattering on low energy Cobalt SE rather than of the Kondo physics, a scattering on a many-body Kondo resonance forming right at the Fermi energy due to correlations on Co orbitals. Though I find results rather interesting I can not unfortunately recommend the manuscript for publication, at least in the present form. I present below few points under question and some remarks.

Answer 2.1: We thank the Reviewer for taking the time of reading our manuscript. We are glad that he/she finds the results interesting. The Reviewer summarized the novelty and originality of our work: our theoretical spectra reproduce very well the experimental ones, which on the basis of our theory evidences that the zero-bias anomalies observed for Co on Cu, Ag, Au(111) surfaces originate from spin-excitations. The physics is then dictated by a relativistic phenomenon, the spin-orbit interaction, which ignites a finite magnetic anisotropy energy favoring the out-of-plane orientation of the magnetic moments. This challenges the mainstream understanding in the community, but gives the opportunity of unravelling new and unexpected physics in what became conventional Kondo physics probed by STM. Motivated by our findings, already several of our experimental colleagues are diving into this type of measurements.

We stress that, at the time of the seminal work of Madhavan et al. on Co/Au(111) [Ref. 5], two possible origins of the zero-bias anomalies observed for Co were discussed. It is our understanding that spin-excitations were not considered as a potential explanation since they were only demonstrated in adatoms 16 years later, in the celebrated work of Heinrich et al. [Ref. 12]. To our knowledge, the experimental evidence that the features observed for Co on the three investigated substrates, Cu, Ag and Au, are really Kondo resonances has not yet been achieved. Our work is thus the outcome of the evolution of this field and the experience collected from important contributions of several groups. We address the concerns of the Reviewer in the following.

1) Novelty: the manuscript follows previous authors’s papers, mainly Ref. 37 (PRB 91, 075405) and Ref. 48 (PRB 89, 235439), extending the calculation of electron self-energies of Ref. 48 to account explicitly for spin-orbit effects which were simulated in Ref. 48 by an auxiliary magnetic field. On my opinion not much new physics is contained in the present work compared to results of Ref. 48 – very similar shapes of magnetic response functions, electron self-energies, and DOS were already reported in Figs. 2, 3 and 5 of Ref. 48 for a Co on Cu(111), together with some other

Prof. Dr. Samir Lounis

Chair of theoretical nanospintronics at RWTH-Aachen
Leader of Functional Nanoscale Structure Probe and
Simulation Laboratory

Tel.: 02461-61-4068

E-Mail: s.lounis@fz-juelich.de

www.fz-juelich.de/pgi/Group-Lounis

PGI-1/IAS-1: Quantum Theory of Materials

cases, the effect of magnetic field was also investigated. I presume that the collective “spinaron” state (introduced in the present manuscript) was also present in Ref. 48 though not discussed explicitly. The authors, on my opinion, should state therefore more clearly if inclusion of spin-orbit effects is crucial and brings any new physics. Or it just corrects for more accurate magnetic anisotropy energy (MAE), related SE energies, etc. with respect to Ref. 48. Also if spin-orbit is accounted for by additional term in the Hamiltonian, like in Eq. 3 of Ref 37, it is controlled anyway by an additional parameter ξ which should be also fixed somehow. The choice of this parameter should be also mentioned and discussed.

Answer 2.2:

i) Refs. 37 and 48 [here, we will keep the previous numbering of the references Phys. Rev. B 91, 075405 (2015) and Phys. Rev. B 89, 235439 (2014), respectively] are the foundations of the work presented in the current manuscript but they are genuinely distinct. The presented new scheme required a reconstruction of our developed framework and emerges from the progress made by generations of PhD students and postdocs. The conventional DFT calculations are obtained from a newly developed Korringa-Kohn-Rostoker scheme, where spin-orbit coupling (SOC) is incorporated in a self-consistent manner equally in the substrate and adatoms [D. S. G. Bauer, PhD dissertation at the RWTH-Aachen (2013)]. In Ref. 48, a possible preliminary scheme to account for a simplified electronic self-energy, obtained with a simplified version of the susceptibility, impacted by spin-excitations is proposed. For instance, a rudimentary projection basis (based on regular scattering solutions that are computed at the Fermi energy) was utilized, which is not that accurate to describe various spin-dynamics properties. However, and more importantly it was lacking the impact of the SOC. The simplest implication of the latter is to open a gap of a few meV in the susceptibility, which has to be precisely described. Ref. 37 shows a scheme to account for SOC in calculating the susceptibility but not in the self-energy. This is subtle in practice since our goal is to describe energy scales of the order of meV range, which requires a careful construction of all the components of the Green function and of the transverse susceptibility, enabled by a powerful projection scheme that provides a very accurate description of the electronic states. This permits a consistent solution of the Dyson-like equation of the dynamical susceptibility, enabling the reliable description of the impact of magnetic anisotropy energies. This was not available in Ref. 48, where a poor’s man approach was utilized to mimic the effect of SOC by applying a magnetic field to open a gap in the susceptibility, which was applied to adatoms on Cu(111) surface. From such an assumption, however, it was/is impossible to make any reasonable claim and systematic interpretation of the experimental data of Co on the three noble metallic substrates (Cu,Ag,Au) since the magnitude of the field is arbitrary and has no physical origin.

ii) Spin-orbit coupling has implications on the susceptibilities that are profoundly different from those induced by merely applying a magnetic field. The main ingredient is the definition of the correct ground state orientation, which can not be known when magnetic fields are used. Additionally, the position and, therefore, the shape of the ZBA are directly determined by the spin-orbit interaction – the correct structure would only be obtained with the use of magnetic fields by chance. The susceptibility’s properties — and therefore the self-energy and the resulting renormalized electronic structure — change dramatically depending on the orientation of the magnetic moment with respect to the lattice. In contrast to the case of SOC-off with a finite magnetic field, the symmetry of the spin-flip relativistic responses, χ^{+-} and χ^{-+} , their shape and weight at

Prof. Dr. Samir Lounis

Chair of theoretical nanospintronics at RWTH-Aachen
Leader of Functional Nanoscale Structure Probe and
Simulation Laboratory
Tel.: 02461-61-4068
E-Mail: s.lounis@fz-juelich.de
www.fz-juelich.de/pgi/Group-Lounis

PGI-1/IAS-1: Quantum Theory of Materials

positive and negative frequencies (imaginary and real part) change dramatically if the moment is lying in an easy-plane or along an easy axis, if it is in the ground state or in a metastable state. This impacts the shape of the self-energies and of the signature of the spin-excitations in the electronic structure.

In the current work, SOC is obtained without adjustable parameters self-consistently in an ab-initio fashion from

$$H_{\text{SOC}} = \frac{1}{(M(r; \epsilon))^2 c^2 r} \frac{\partial V(r)}{\partial r} \mathbf{L} \cdot \mathbf{S},$$

where $V(r)$ is the ab-initio potential and the relativistic mass

$$M(r; \epsilon) = \frac{1}{2} - \frac{\epsilon - V(r)}{2c^2}$$

and c being the speed of light.

To come to the conclusions of our paper, which goes against the main-stream interpretation of the ZBA, we followed a rigorous scientific approach by accounting for SOC in the MBPT approach, which is done for the first time in the current paper. In the ab-initio community, this is a huge leap forward and a *tour de force* (both analytically and computationally) that allowed us to perform the calculations systematically on the three surfaces without any adjustable parameters. It is only when armed with the evidences shown in the paper that we could propose a completely different and challenging interpretation for the zero-bias anomalies observed for Co on Cu, Ag and Au(111) surfaces. While the physics is dictated by the magnetic anisotropy energy, which is of a few meV and thus not trivial to evaluate, we demonstrate a one-to-one almost perfect agreement to the available experimental data. Our work and its conclusions go clearly far beyond the works mentioned by the Reviewer.

iii) Concerning the way SOI is accounted in formalism, we would like to highlight that we already mention in the Methods section “Our first-principles approach is implemented in the framework of the scalar-relativistic full-electron Korringa-Kohn-Rostoker (KKR) Green function augmented self-consistently with spin-orbit interaction [35,36], where spin-excitations are described in a formalism based on time-dependent density functional theory (TD-DFT) [29,37,38] including spin-orbit interaction.”

We also introduced new changes to make this point more clear:

“Our first-principles simulations indicate that the observed features find their origin in inelastic spin-excitations (SE), as known for other systems [12,27–34], which are gaped SE owing to the magnetic anisotropy energy that favors the out-of-plane orientation of the Co moment. **Therefore, the physics is dictated by relativistic effects introduced by the spin-orbit interaction.**”

“The spinaron is characterized by an energy and a lifetime (**settled by $\text{Im}(G\Sigma)$**), both **affected** by the spin-orbit interaction, since it dictates the magnitude of the SE-gap **defining the self-energy**, and the electron-hole excitations. ”

Prof. Dr. Samir Lounis

Chair of theoretical nanospintronics at RWTH-Aachen
Leader of Functional Nanoscale Structure Probe and
Simulation Laboratory

Tel.: 02461-61-4068

E-Mail: s.lounis@fz-juelich.de

www.fz-juelich.de/pgi/Group-Lounis

PGI-1/IAS-1: Quantum Theory of Materials

2) Novelty and reliability: As far as I know the nature of ZBA in conductance has a rather long discussion in the surface science community, there are many cases where these features, ascribed initially to Kondo-related physics, were then attributed to some inelastic processes (mainly to a scattering on low energy phonons but also on magnetic excitations), so it is still under discussion. On the other side, the Co ad-atoms represent the most traditional case where the community seems to converge to the Kondo-related interpretation. I think therefore that the authors should be more modest and present their results as another indication towards non-Kondo interpretation, not denying however completely the Kondo-related realization. The two effects, Kondo screening and inelastic electron/spin scattering, could possibly even co-exist.. If the authors really want to discard the Kondo interpretation, they should provide some more arguments. For example, the spin $S > 1/2$ of Cobalt (it is $S=1$) together with finite MAE could prevent for the Kondo screening of the Cobalt spin. In this respect, I would strongly suggest to look at the Cu(001) surface (and other (001) surfaces) where very different (from presented here (111) orientations) line-shapes were reported experimentally (Ref. 11). If Kondo fluctuations for Co are frozed by above arguments, the presented here “spin excitation” hypothesis should also explain and reproduce experimental asymmetric line-shapes.

Answer 2.3: We thank very much the reviewer for pointing out the aspect related to the large spin of Co ($> 1/2$) and its finite MAE, which reinforce our interpretation of the results. The argument of the reviewer becomes stronger by noticing that the MAE favors the out-of-plane orientation of the magnetic moment. We followed his/her advice and added the following note to the manuscript, which reinforces our interpretation of the results: “Considering that the large magnetic moments of the Co adatoms are characterized by an out-of-plane easy axis, Kondo-screening is unlikely to occur [21], and enforces the view that the zero-bias anomalies result from spin-excitations.”

To our knowledge the ZBA of Co deposited on Cu, Ag and Au were not under debate before submission of our work. With the suggestion of the Reviewer, we looked at other examples where Kondo interpretation of ZBA was challenged. We found some in the context of quantum wires, which we now cite in the manuscript. We amended the following text to the Discussion section: “We note that there are other examples of materials, such as quantum wires, where zero-bias anomalies have been challenged to be Kondo features [50,51]. However, the adatoms investigated in the current work represent the most traditional systems, where the surface science community converges to the Kondo-related interpretation.”

We would like to add that we certainly do not deny the existence of Kondo-related interpretation but we cannot refute the nice agreement between our ab-initio theory and the various experimental data for Co adatoms on Cu, Ag and Au(111) surfaces. Overall, we trust that we were very modest in our description of our results and their possible implications. We mentioned at various places, to the best of our knowledge, the contributions of our predecessors in this field and were very careful about the outcome of our work. It is in particular quite exciting to realize that our work shows that the seminal work of Madhavan et al. already identified an electronic signal permitting to measure the MAE of Co atoms on Au(111) in 1998, prior to the very nice XMCD measurements of our colleagues Gambardella and coworkers on Co/Pt(111) surface in 2003 [Science 300,

Prof. Dr. Samir Lounis

Chair of theoretical nanospintronics at RWTH-Aachen
Leader of Functional Nanoscale Structure Probe and
Simulation Laboratory
Tel.: 02461-61-4068
E-Mail: s.lounis@fz-juelich.de
www.fz-juelich.de/pgi/Group-Lounis

PGI-1/IAS-1: Quantum Theory of Materials

1130 (2003)]. The goal of our work is to motivate further studies in this field and ignite further experiments, which could lead to a better understanding of the ZBA. Several members of this community were excited about our results, which undoubtedly pushed the realization of the presented work.

Motivated by the suggestion of the Reviewer, we performed additional calculations of Co adatoms on Cu(001) and Ag(001) surfaces as well as Ti adatom on Ag(001). A one-to-one comparison with the experimental data is provided in the Supplement Figure 1 and in Figure 2. One sees the compelling agreement between theory and experiment providing an additional evidence that on the (001) surfaces, spin-excitations are probably at play in the realization of the ZBA. The feature obtained for Ti is sharper than the one measured experimentally. We stress that these results, as already noticed by the Reviewer can be affected by the magnitude of the MAE and the amount of electron-hole excitations responsible for the damping of the spin-excitations. This definitely motivates further studies on these particular systems and other ones. The following text has been added to the Discussion section: “Additional simulations performed on Co adatoms on Cu and Ag(001) surfaces as well as Ti adatom on Ag(001), shown in Supplementary Figure 1, provide an additional evidence that spin-excitations are potentially present on other materials, giving rise to the experimentally observed zero-bias anomalies.”

3) I find the discussion on “spinaron” rather superficial, could the authors provide a more detailed description of what it is and how do they find it. Probably they could provide a figure with real and imaginary parts of self-energy? Also I would appreciate some more details on its energies and life times for different substrates. I wonder also if there is any relation between “spinaron” and Kondo resonance since they are both collective states made of conduction electron+local spin. Can “spinaron” be considered as a kind of “approximate” Kondo state approached within mean-field DFT method? Does the “spinaron” exist also if MAE is zero (gapless spin excitations) or it needs a finite anisotropy (gapped SE)?

Answer 2.4: We thank the reviewer for raising very interesting questions. Following his/her advice, we improved our discussion on the origin of the spinaron (see also Answer 1.3 to the first Reviewer). The spinaron is a bound state emerging from the interaction of the spin-excitation and the electron, which is encoded in the electronic self-energy. After solving the Dyson equation to obtain the renormalized Green function

$$\begin{aligned}\underline{G}_R(\epsilon) &= [1 - \underline{G}(\epsilon) \underline{\Sigma}(\epsilon)]^{-1} \underline{G}(\epsilon) \\ &= [\underline{G}(\epsilon)^{-1} - \underline{\Sigma}(\epsilon)]^{-1},\end{aligned}$$

it is only when the real part of the inverse of the Green function intersects the real part of the self-energy that a localized state can emerge. To be more precise, this happens when the real part of $\underline{G}(\epsilon)\underline{\Sigma}(\epsilon) = 1$, meaning that $\text{Re}[\underline{G}(\epsilon)]\text{Re}[\underline{\Sigma}(\epsilon)] - \text{Im}[\underline{G}(\epsilon)]\text{Im}[\underline{\Sigma}(\epsilon)] = 1$. So in general both the real and imaginary parts of the self-energy can be involved in realizing the spinaron. Moreover, it is important to keep in mind that we are dealing with a multi-orbital problem. Note that the lifetime is defined by the imaginary part of $\underline{G}(\epsilon)\underline{\Sigma}(\epsilon)$. The additional interaction introduced by the self-energy can act as an attractive potential that can localize electronic states in a given energy window. The energy and lifetime ($\epsilon_{\text{spinaron}}, \tau_{\text{spinaron}}$) of the spinaron, as obtained from the theoretical spectra in the vacuum, on the three surfaces (listed now in the main text) are: (4.42 meV, 0.34 ps), (12.6 meV, 0.20 ps) and (9.41 meV, 0.20 ps) for Cu, Ag and Au(111), respectively. In the cases shown in the

Prof. Dr. Samir Lounis

Chair of theoretical nanospintronics at RWTH-Aachen
Leader of Functional Nanoscale Structure Probe and
Simulation Laboratory

Tel.: 02461-61-4068

E-Mail: s.lounis@fz-juelich.de

www.fz-juelich.de/pgi/Group-Lounis

Figure 2: Zero-bias anomalies calculated and measured on Co adatoms on Cu and Ag(001) surfaces as well as Ti adatom on Ag(001) surface. For Co/Cu(001), the agreement with the experimental data [1,2] is not as perfect as it is on the (111) surfaces. We recover, however, the observed step-like behavior. A reason could be the underestimation of the magnetic anisotropy energy of Co on Cu(001), which can shift slightly the spectrum around. For Co/Ag(001), the agreement is rather good when compared to the data of Ternes et al. [3]. It is interesting to notice, however, that the measurements of Wahl et al. [4] lead to different spectra. The disagreement between the two experimental data can be surprising at first sight. We conjecture that this can be induced by the presence of hydrogen or by the difference in the probing tip, which can change the shape of the features. This strongly motivates further experimental investigations. The case of Ti/Ag(001) measured by Nagaoka et al. [5] is rather reasonably described by our theory. Our zero-bias anomaly is sharper than the experimental one. Overall, the agreement between theory and experiment is rather good on the (001) surfaces, which indicates that spin-excitations, as discussed in the main text, is a plausible origin of the low-energy features on the (001) surfaces of Cu and Ag, similarly to the (111) surfaces. Note that the position of the theoretically obtained features hinges on the ability to evaluate the magnetic anisotropy energy of the adatoms, which is not always trivial. The experimental data adapted with permission from Refs. 1-5. Copyright (2002), (2004) and (2007) by the American Physical Society.

manuscript, one can identify the spinaron by performing a one-to-one comparison between the spin-resolved self-energy and the local density of states. The intrinsic spin-excitations lead to the steps observed in the imaginary part of the self-energies. The spinaron shows up only in the renormalized local density of states. As shown in the orbital-resolved adatom LDOS (Figure 3 and Supplementary Figure 5), the spinaron emerges from the majority-spin d_{z^2} orbital. Another possibility to identify the origin of the spinaron is to track when

Prof. Dr. Samir Lounis

Chair of theoretical nanospintronics at RWTH-Aachen
Leader of Functional Nanoscale Structure Probe and
Simulation Laboratory

Tel.: 02461-61-4068

E-Mail: s.lounis@fz-juelich.de

www.fz-juelich.de/pgi/Group-Lounis

Figure 3: **Orbital-resolved renormalized local density of states of Co adatoms on the three surfaces Cu, Ag, Au(111).** The spinaron emerges from the majority-spin d_{z^2} orbital. **[Filipe: please, check this caption here and in the supplement]**

$\text{Re}[\underline{G}(\epsilon)]\text{Re}[\underline{\Sigma}(\epsilon)] - \text{Im}[\underline{G}(\epsilon)]\text{Im}[\underline{\Sigma}(\epsilon)] = 1$, which can be performed approximately for each orbital. This is shown in Figure 4 (Supplementary Figure 6) for the specific case of Co/Cu(111), where one sees that the inverse of the majority-spin $\text{Re}[\underline{G}(\epsilon)]$ of d_{z^2} -nature intersect $\text{Re}[\underline{\Sigma}(\epsilon)]$. The contribution of $\text{Im}[\underline{G}(\epsilon)]\text{Im}[\underline{\Sigma}(\epsilon)]$ (and other interference effects) shifts the spinaron to lower energies. In the Supplementary Material, we amended curves showing where the real part of the Green function bisect the real part of the self-energy, which is the essential ingredient for the realization of the spinaron.

The text added to the manuscript is: “One can recognize it (spinaron) either from a one-to-one comparison between the spin-resolved LDOS and the self-energy, as being a feature not present in the latter one (see Supplementary Figures 3 and 4) or from tracking the intersections of Green functions and self-energies leading to the vanishing of the denominator of Eq. 1. The presence of spin-fluctuations affect the electronic behavior in terms of the electron-SE interaction encoded in the self-energy. This additional interaction can act as an attractive potential permitting the localization of electrons in a finite energy window, giving rise to a bound state. The spinaron emerges then when the denominator of the Dyson equation, Eq. 1, cancels out, i.e. when $\text{Re}(\underline{G}\underline{\Sigma}) = 1$, which occurs for the d_{z^2} orbital having the ideal symmetry to be detected by STS, as illustrated in Supplementary Figures 5 and 6.”

“Similarly to Au, the spinaron originates from the d_{z^2} on Ag and Cu, conferring the right symmetry to be detected efficiently with STS (more details are provided in Supplementary Figures 5 and 6)”.

“The energies and lifetimes ($\epsilon_{\text{spinaron}}, \tau_{\text{spinaron}}$) of the spinarons as obtained from the theoretical spectra in vacuum are: (4.42 meV, 0.34 ps), (12.6 meV, 0.20 ps) and (9.41 meV, 0.20 ps) for Cu, Ag and Au(111), respectively, which are of the same order of magnitude than those of the intrinsic spin-excitations listed in Table 1.”

Prof. Dr. Samir Lounis

Chair of theoretical nanospintronics at RWTH-Aachen
Leader of Functional Nanoscale Structure Probe and
Simulation Laboratory

Tel.: 02461-61-4068

E-Mail: s.lounis@fz-juelich.de

www.fz-juelich.de/pgi/Group-Lounis

Figure 4: **Orbital-resolved real part of inverse Green function and self-energy of the majority spin channel of Co adatoms on Cu(111) surface.** The intersection occurring in the d_{z^2} -orbital leads to a vanishing denominator of the Eq 1 of the main text and, therefore, to the spinaron. The contribution of $\text{Im}[\underline{G}(\varepsilon)]\text{Im}[\underline{\Sigma}(\varepsilon)]$ (and other interference effects) shifts the spinaron to lower energies.

We also refer to the changes listed in Answer 1.3 to the first Reviewer.

In the Supplementary information, we added a comparison of the real and imaginary parts of the self-energies (see Figure 5, which is now added as Supplementary Figure 3). The spectra shown in the vacuum are resulting from the decay of the electronic states of the adatoms, which, for completeness, are also shown as Fig. 6 and added as Supplementary Figure 4.

The question on the relation between the “spinaron” and Kondo resonance is an exciting problem, which we are pursuing. Both phenomena result from the dynamical coupling of the spin moment to the bath of electrons. The spinaron has a spin-character. It can be realized in one or in the spin-channels. For Co on Cu, Ag, and Au(111) surface, it emerges in the majority-spin channel but not in the minority-spin one. In general, the spinaron does not need to live in-between the intrinsic spin-excitations, or on top of one of them. It can also be outside the spin-excitation “effective gap” since its existence is related to bisection of the real-part of the self-energy and the real part of the inverse of the Green function. The emergence of the spinaron is thus not even necessarily located at low energies. However, when this is the case, the spinaron has a longer lifetime and its experimental detection may be achieved. With these arguments, we refrain from identifying the spinaron as an approximate Kondo state.

Regarding the MAE, we did not demonstrate so far a clear correlation between the spinaron and the magnitude of the MAE because of the complexity of the real materials that we are investigating, where we deal with

Prof. Dr. Samir Lounis

Chair of theoretical nanospintronics at RWTH-Aachen
Leader of Functional Nanoscale Structure Probe and
Simulation Laboratory

Tel.: 02461-61-4068

E-Mail: s.lounis@fz-juelich.de

www.fz-juelich.de/pgi/Group-Lounis

Figure 5: **Comparison of the spin-resolved imaginary and real parts of the trace of the Co adatoms self-energies on the three surfaces Cu, Ag, Au(111).** Similarly to Au(111) surface (shown also in the main Figure 2b), the steps characterizing the imaginary part of the self-energies are asymmetric. The one generated at negative bias voltage corresponding to the majority-spin channel is the largest on Ag. This is induced by the large minority-spin local density of states (see Supplemental Figure 2), which defines the height of the step.

Figure 6: **Comparison of the renormalized local density of states of Co adatoms on the three surfaces Cu, Ag, Au(111).** The adatoms electronic states decay in the vacuum theoretical spectra obtained in the vacuum. Therefore the shape of the signature of the spin-excitations and of the spinaron do not have to be the same in the adatom and in the vacuum.

a multi-orbital problem. The MAE dictates the position of the intrinsic spin-excitations. Their broadening is shaped by the electron-hole excitations defined by the electronic structure of the adatom. Increasing the MAE opens the effective spin-excitation gap, which is a positive ingredient enabling the observation of a spinaron. However, at larger energies, the electron-hole excitations are more important, inducing a broadening of the

Prof. Dr. Samir Lounis

Chair of theoretical nanospintronics at RWTH-Aachen
Leader of Functional Nanoscale Structure Probe and
Simulation Laboratory
Tel.: 02461-61-4068
E-Mail: s.lounis@fz-juelich.de
www.fz-juelich.de/pgi/Group-Lounis

PGI-1/IAS-1: Quantum Theory of Materials

intrinsic spin-excitations and of the corresponding self-energies. This can hinder the realization of the spinaron by shifting and broadening it. Reducing the MAE, increases the lifetime of the spin-excitations and should in principle favor the observation of a spinaron, at the condition that the electronic states of the adatoms permits it.

As probably noticed by the Reviewer, this work is already opening very interesting lines of research and as indicated by the first Reviewer, it gives fresh air in this field of research with challenging novel views on fundamental quantum features. We are convinced that this will trigger more activities in this field of research.

4) Could the authors explain in more detail how do they construct vacuum DOS, at which distance above the surface, what is orbital composition of it (s, p, d)? Why in Fig. 2d the authors present the changes of LDOS but not the total ones which I think should be probed by an STM tip? Moreover, in Fig. 3b it seems that the solid lines are not the averages, they do not lie in between dashed lines (up and down).

Answer 2.5: (i) The vacuum in our formalism is divided into Voronoi cells, in which an orbital decomposition can be performed but we are plotting the total DOS. The orbital decomposition is not meaningful in the vacuum. The vacuum LDOS is obtained at a distance of 6.3 Å above the adatom for the Cu(111) surface and 7.1 Å for the Ag and Au(111) ones. We added the latter information into the main text: “here assumed to be located at 6.3 Å above the adatom for the Cu(111) surface and 7.1 Å for the Ag and Au(111) ones.”

(ii) Experimentally, the tunneling spectra are in general renormalized by subtracting the background spectra to remove tip-related electronic features or substrate (see e.g. Refs. [Science 280, 567–569 (1998), Phys. Rev. Lett. 93, 176603 (2004), Jpn. J. Appl. Phys. 44, 5328–5331 (2005), Review of Scientific Instruments 79, 043104 (2008)]). In our theory, we use the Tersoff-approximation, which makes a specific assumption on the tip nature and how it contributes to the signal. A way of removing the contribution of the tip in a systematic way is to renormalize the LDOS by that of the defect-free surface, which is quasi identical to Δn plotted in the main manuscript.

(iii) Indeed, in Fig. 3b, the solid lines are not the averages. They are the sum of the dashed lines corresponding to each spin-channel. They are moved, since plotted in arbitrary units, such that the various structures can be seen and identified from their location and spin channel. We improved the description of the figure in the caption of the figure: “The total (solid) signal is obtained by the sum of both spin channels, which were shifted for a better visualization and comparison.”

5) I wonder also about the effect of possible “Hubbard” +U corrections. In static DFT calculation U usually moves significantly d-states from the Fermi energy (occupied states down and empty up in energy). Since the self-energies are the convolutions of DFT d-states DOS and response function, I would expect that d-DOS modification at the Fermi energy due to +U corrections would also affect importantly the results.

Prof. Dr. Samir Lounis

Chair of theoretical nanospintronics at RWTH-Aachen
Leader of Functional Nanoscale Structure Probe and
Simulation Laboratory

Tel.: 02461-61-4068

E-Mail: s.lounis@fz-juelich.de

www.fz-juelich.de/pgi/Group-Lounis

PGI-1/IAS-1: Quantum Theory of Materials

Answer 2.6: The self-energy that we calculate can be considered as an effective dynamical U obtained in an ab-initio manner. Adding a U the way the reviewer is suggesting would lead to the following: (i) the electronic states, in particular the ones at the Fermi energy, will change, and the exchange splitting will increase. (ii) the MAE will change. Item (i) can modify the magnitude of the electron-hole excitations and affect the lifetime of the spin-excitations while (ii) can modify the location/energy of the spin-excitations. So, to explore the impact of U , one has to choose values that do not deteriorate the location/energy of the spin-excitations, which seems to be nicely described so far. Calculations with U are often used in practice to fit experimental data, but this is not needed here. Such investigations go beyond the scope of the current work and requires important method/code developments to account for U in our implementations of spin-dependent TD-DFT and MBPT. The latter is being developed but requires one more generation of PhD students.

Prof. Dr. Samir Lounis

Chair of theoretical nanospintronics at RWTH-Aachen
Leader of Functional Nanoscale Structure Probe and
Simulation Laboratory

Tel.: 02461-61-4068

E-Mail: s.lounis@fz-juelich.de

www.fz-juelich.de/pgi/Group-Lounis

PGI-1/IAS-1: Quantum Theory of Materials

Third Reviewer

This paper reports on tunneling spectra taken by scanning tunneling microscopy (STM) on single Co atoms adsorbed on noble metal (111) surfaces. Whereas those spectra have been so far interpreted as a Fano resonance of the Kondo resonance, the present paper claims they are explained with inelastic tunneling (spin-flipping) and the interaction of the excited spin with substrate electrons based on the authors' theoretical calculations. The presented agreements with experimental results are good, but indeed it was surprising to me whether and why this kind of rather simple interpretation really works. If this is really true several questions arise as mentioned below, and I would like to judge whether the paper should be accepted or not in Nature Communications after the questions are clearly solved.

Answer 3.1: We thank the Reviewer for reading and assessing our work. Indeed, the results obtained with our all-electron first-principles methodology leads to a nearly perfect systematic description of the zero-bias anomalies of Co adatom on Cu, Ag and Au(111) surfaces. As the Reviewer highlights, the interpretation is simple but lies on solid theoretical foundations that requires intensive ab-initio simulations that incorporates spin-flip excitations including the impact of the spin-orbit interaction. The final spectra is obtained via a scheme based on multiple-scattering theory implemented within time-dependent density functional theory and interfaced with many-body perturbation theory.

In this paper, the authors claim quite large magnetic anisotropic energy (MAE) in these system. This is quite surprising to me because so far only Co/Pt(111) system is known to have large MAE. As authors mentioned in this manuscript, in order to have MAE spin-orbit interaction (SOI) plays a crucial role (2nd order perturbation). It is then strange to me why the three substrates in particular Cu and Au exhibits similar values of MAE. Au should have much larger SOI and thus MAE as a heavy element. The spin relaxation time by Fermi sea should also be dependent on SOI but three substrates give similar lifetime values. These seem strange to me.

Answer 3.2: The MAE of Co on Cu, Ag and Au(111) are reasonable. The physics of MAE is rich, depending on many factors, and the recipe for large MAE is not that trivial. Having a heavy substrate does not necessarily lead to a large MAE. The reason is that the electronic structure of the adsorbate matters. It can change depending on the relaxation of the adatoms, the electronic structure of the bare substrate and the hybridization strength between the electronic states of the adatom and of the substrate. We note, for instance, that the record of largest MAE is taken by MgO/Ag(001) [Science 329, 1628 (2010)] and not Pt(111). Pt is indeed a heavy element with a large SOC, which should favor the probability of finding elements with large MAE. One should not however underestimate the impact of the electronic structure of the adatom itself impacted by the crystal field splitting and hybridization mechanisms. For instance, the polarization cloud of Pt affects non-trivially the magnitude and sign of the MAE [PRL 111 (15), 157204 (2013)], which modifies strongly the intuitive picture we have/expect for Pt as a substrate. For example, the same Co atom can have very different MAE depending on its stacking site, being fcc or hcp on the 111 surface of Pt [PRB 94 (12), 125402 (2016)]. The work of Siper et al. [J. Phys.: Condens. Matter 26 196002 (2014)] devoted to the MAE of Co adatoms on

Prof. Dr. Samir Lounis

Chair of theoretical nanospintronics at RWTH-Aachen
Leader of Functional Nanoscale Structure Probe and
Simulation Laboratory

Tel.: 02461-61-4068

E-Mail: s.lounis@fz-juelich.de

www.fz-juelich.de/pgi/Group-Lounis

PGI-1/IAS-1: Quantum Theory of Materials

various substrates, among them Cu, Ag and Au(111) surfaces demonstrated that the contribution of spin-orbit interaction of those substrates is negligible with respect to the effect of the spin-orbit interaction of Co. This is assessed in their ab-initio simulations by switching on and off the SOI of the substrate and of the adatom. Note that their MAE are a factor of 3 larger than ours, i.e. larger than that of Pt, which is probably induced by their neglect of geometrical relaxations. This behavior can be understood from perturbation theory a la Bruno [Phys. Rev. B 39, 865 (1989)] is given by the spin-orbit coupling strength of Co multiplying the change of the orbital moment upon the rotation of the spin moment. Both ingredients are large for Co on the three studied surfaces, which leads to a finite MAE. So the electronic structure of Co on Cu and on Au are very similar, which not only leads to a similar MAE but also to similar orbital moments (shown in Table 1 of the main text).

The authors mention inelastic spin flip energy is given as $4(\text{MAE})/M_{\text{spin}}$. It is also strange to me why it is not simply MAE. Why is M_{spin} involved here?

Answer 3.3: This is a standard result known from the susceptibilities obtained from linear response theory of a Heisenberg model (see e.g. Ref. [M. dos Santos Dias, Phys. Rev. B 91, 075405 (2015)]). In reality it is bit more complicated, if one accounts for the impact of electron-hole modes and the consequent finite lifetimes of the excitations. We give here a simple interpretation of this formula. The Larmor frequency defining the spin-flip energy induced by the presence of a magnetic field, B , is given by $gB = 2B$. The effective field corresponding to the $\text{MAE} = Ke_z^2$, where e_z is the unit vector of the magnetic moment, is given by $B_{\text{MAE}} = \frac{\partial \text{MAE}}{\partial M} = 2\frac{K}{M}$. Therefore the spin-flip energy is given by $2g\frac{K}{M} \approx 4\frac{K}{M}$.

It have been known that single Co atoms adsorbed on Pt(111) substrate show a large MAE. Spectra showing inelastic tunneling were reported, but, not Kondo-like features, which seems inconsistent with the model presented in this manuscript. What is the difference between the three substrates and Pt, and how the difference is induced?

Answer 3.4: Our results indicate that spin-flip excitations lead to low-energy features matching nicely those observed for Co adatoms on Cu, Ag and Au(111) surfaces. Since Co is magnetic on Pt(111) with a large MAE also favoring an out-of-plane orientation of the spin moment, we expect similar physics, i.e. the occurrence of spin-flip excitations. Because of the larger MAE on Pt, the inelastic features would occur at larger energies than on Cu, Ag and Au(111). So this is all consistent with the message conveyed in our manuscript. This leads necessarily to a larger broadening of the features, making the signal more difficult to observe. Note also that the electronic states involved in the excitations process need to have the right symmetry to ensure slower decay into the vacuum to be detected by the STM tip.

Whether the model is correct or not can be easily checked experimentally, by measuring temperature dependence of the spectra, or performing spin-polarized tunneling spectroscopy on the adatoms. I wonder why authors do not collaborate with experimental groups. Since most of the Kondo resonance works were done at 4K so far, performing at lower temperature is enough to distinguish the two interpretations, I guess.

Prof. Dr. Samir Lounis

Chair of theoretical nanospintronics at RWTH-Aachen
Leader of Functional Nanoscale Structure Probe and
Simulation Laboratory

Tel.: 02461-61-4068

E-Mail: s.lounis@fz-juelich.de

www.fz-juelich.de/pgi/Group-Lounis

PGI-1/IAS-1: Quantum Theory of Materials

Answer 3.5: We share the enthusiasm of the Reviewer on the implications of our results and we agree that this is an exciting problem to check experimentally. Because of the large broadening of the zero-bias anomalies, it is not straightforward to track the reaction of the features to different stimuli. We fully agree that considering spin-flip excitations as a simple and reliable mechanism for the observed features should make it easier to guide experimental groups to verify our interpretations. Therefore, the importance of our results to provide a solid foundation for those difficult studies. It is certainly the excitement of our experimental colleagues that pushed us to develop the methodology used here and carry out the heavy ab-initio investigations. From their feedback, we are confident that this work opens new avenues of research in this field, rising new questions and interesting physics to explore. We are aware of various investigations currently performed to explore the various aspects highlighted in our study.

In the manuscript, the authors provide rather detailed explanations on generally known things, such as energy shift/broadening of self-energy, renormalization of Green's function etc. On the other hand, physical explanations on the calculated results, such as why the three substrate has different spectra, what are the spin states and orbital occupation of adsorbed Co atoms, why background slopes of the three spectra are different (Cu and Au go up with the voltage but Ag goes down) etc. are missing. These are quite important information for understanding the physics and interpretation of the model and should not be missed.

Answer 3.6: We thank the Reviewer for his valuable suggestions. The spin and orbital moments are listed in Table 1. The aspect related to the difference between the spectra on the three substrates is addressed in Answers 1.3, 2.4 to the other Reviewers, where we also listed the amendments made to the manuscript.

Further changes made on the manuscript:

- Initial sentence of the paper: "Signatures of many-body phenomena in solid state physics are diverse [1-5]. One of them is the Kondo effect emerging from the interaction between the sea of electrons in a metal and the magnetic moment of an atom [6,7], whose signature is expected below a characteristic Kondo temperature T_K ."
- In the previous convention, we used $\Gamma = 2\hbar/(\text{FWHM})$, while now we use $\Gamma = \hbar/2(\text{FWHM})$, in agreement with the convention used in Ref. [A. A. Khajetoorians et al., Phys. Rev. Lett. 111, 157204 (2013)]. Therefore, the results on Table 1 were divided by 4.
- In the Discussion section: "..., which stimulates further theoretical developments permitting the ab-initio investigation of Kondo features, spinarons, spin-excitations and spin-orbit driven physics on equal footing" followed by "...and unravel the complexity and richness of the physics behind the spinaron."
- Added titles for different sections and sub-sections.

REVIEWER COMMENTS

Reviewer #1 (Remarks to the Author):

Bouaziz et al. addressed my remarks properly. I do recommend publication of the current manuscript to nature communications.

Reviewer #2 (Remarks to the Author):

I am satisfied overall with detailed and thorough answers of the authors to my questions with additional calculations on (001) surfaces.

I think that the manuscript is now in a good shape for publication with few optional comments which I suggest the authors to address.

Minor point:

My Remark 1 concerning the novelty with respect to the author's previous papers.

I agree that the rigorous inclusion of spin-orbit interactions -- which are at the origin of magnetic anisotropy -- into the developed early theoretical approach represents an important step allowing to analyze, in a free-parameter manner and on the same footing,

Co ad-atoms on different substrate. Clearly, this implementation was not straightforward and required a lot of time and effort.

But what I meant was that the crucial physics allowing to compare well with experimental tunneling spectra --

gapped spin-excitations and the spinaron -- was already approached in a previous simplified scheme where magnetic anisotropy was modelled with auxiliary magnetic field. Therefore I still suggest to add few more sentences to compare more clearly with the previous method and the results.

Major point:

More important point concerns my Remark 5 about possible effects of Hubbard-like correction U on Co d-orbitals.

Usually for such low-coordinated geometries some finite U , ranging normally from 2 and 5 eV, is used in DFT calculations, its exact value can be indeed obtained by fitting to known experimental results (or calculated

using some linear response techniques). I think that the argument that with $U=0$ the calculated results reproduce well the experiment

while the finite U would probably deteriorate the comparison does not seem to be very solid.

On the other side, the finite U in conjunction with the crystal field, will favour asymmetric occupation

of minority spin Co d-orbitals with three of them fully occupied and two -- empty, which is quite different

to rather symmetric (and partial) occupation of all five d-orbitals as presented by authors in Fig. 1 of Response.

This could be even the case of $U=0$ as it was reported in and Ref. 45 (PRB 97, 235442 (2018)) and in PRB 92, 045119 (2015) for Co on Cu surfaces.

Altogether that will affect (essentially reduce) the Co DOS at the Fermi energy and as a consequence is expected to alter many of presented

in the paper results as fairly admitted by authors.

Of course, I agree, that this problem goes somewhat beyond the scope of the manuscript but

I would like the authors to be aware of this problem and I would suggest them to add a small discussion

about this point in the text.

Reviewer #3 (Remarks to the Author):

Due to the revisions and replies made by the authors I think I understand better at least than before, for which I appreciate the authors' efforts.

The authors demonstrate in this manuscript that the tunneling spectra taken on Co adsorbates on noble metal (111) surfaces, which have been believed Kondo resonance in surface science community, can be explained with a gap due to spin-flipping inelastic tunneling plus a bound state called spanaron. The scenario the authors propose could be true, but it seems to me dangerous at the present stage to publish it in the prestigious journal and to provide a kind of scientific approval to the community.

One of the major reasons of the my hesitate about the acceptance is lack of experimental supports. The authors nicely fit the spectra with the their model, but the spectra were also nicely explained with the Kondo model. According to the authors' calculation results, Co adatoms exhibit out-of-plane magnetization, which is obviously incompatible with the Kondo model. The presence of the magnetization can be detected by using spin-polarized STM presumably without difficulty. XMCD may also can be used for the detection.

Whereas the spectral feature is "a dip" for the Kondo model, it is "a gap" for the authors model. The temperature broadening thus should behave quite different from each other. Taking temperature-dependent tunneling spectra, which is not very difficult using advanced commercial STM setups, will provide a clear answer which is plausible.

It seems the authors already contacted with some leading and pioneering researchers in this field. I wonder why they do not provide experimental results supporting the scenario. If experimental results inconsistent with the Kondo model are presented it will be immediately accepted without hesitation.

Prof. Dr. Samir Lounis

Chair of theoretical nanospintronics at RWTH-Aachen
Leader of Functional Nanoscale Structure Probe and
Simulation Laboratory
Tel.: 02461-61-4068
E-Mail: s.lounis@fz-juelich.de
www.fz-juelich.de/pgi/Group-Lounis

PGI-1/IAS-1: Quantum Theory of Materials

First Reviewer

Bouaziz et al. addressed my remarks properly. I do recommend publication of the current manuscript to nature communications.

Answer 1.1: We thank once more the Reviewer for his/her recommendation to publish our manuscript.

Second Reviewer

I am satisfied overall with detailed and thorough answers of the authors to my questions with additional calculations on (001) surfaces. I think that the manuscript is now in a good shape for publication with few optional comments which I suggest the authors to address.

Minor point: My Remark 1 concerning the novelty with respect to the author's previous papers. I agree that the rigorous inclusion of spin-orbit interactions – which are at the origin of magnetic anisotropy – into the developed early theoretical approach represents an important step allowing to analyze, in a free-parameter manner and on the same footing, Co ad-atoms on different substrate. Clearly, this implementation was not straightforward and required a lot of time and effort. But what I meant was that the crucial physics allowing to compare well with experimental tunneling spectra – gapped spin-excitations and the spinaron – was already approached in a previous simplified scheme where magnetic anisotropy was modelled with auxiliary magnetic field. Therefore I still suggest to add few more sentences to compare more clearly with the previous method and the results.

Major point: More important point concerns my Remark 5 about possible effects of Hubbard-like correction U on Co d-orbitals. Usually for such low-coordinated geometries some finite U , ranging normally from 2 and 5 eV, is used in DFT calculations, its exact value can be indeed obtained by fitting to known experimental results (or calculated using some linear response techniques). I think that the argument that with $U=0$ the calculated results reproduce well the experiment while the finite U would probably deteriorate the comparison does not seem to be very solid. On the other side, the finite U in conjunction with the crystal field, will favour asymmetric occupation of minority spin Co d-orbitals with three of them fully occupied and two – empty, which is quite different to rather symmetric (and partial) occupation of all five d-orbitals as presented by authors in Fig. 1 of Response. This could be even the case of $U=0$ as it was reported in and Ref. 45 (PRB 97, 235442 (2018)) and in PRB 92, 045119 (2015) for Co on Cu surfaces. Altogether that will affect (essentially reduce) the Co DOS at the Fermi energy and as a consequence is expected to alter many of presented in the paper results as fairly admitted by authors. Of course, I agree, that this problem goes somewhat beyond the scope of the manuscript but I would like the authors to be aware of this problem and I would suggest them to add a small discussion about this point in the text.

Answer 2.1: We thank the Reviewer for appreciating the new version of the manuscript and for recommending

Prof. Dr. Samir Lounis

Chair of theoretical nanospintronics at RWTH-Aachen
Leader of Functional Nanoscale Structure Probe and
Simulation Laboratory
Tel.: 02461-61-4068
E-Mail: s.lounis@fz-juelich.de
www.fz-juelich.de/pgi/Group-Lounis

its publication. We considered the optional comments and decided to include the following sentence in subsection "Systematic study of Co adatoms on Cu, Ag, Au(111) surfaces": "Interestingly, the spectra obtained for Co/Cu(111) are in line with those reported in Ref. [47] based on a simplified theoretical approach (more details are provided in Supplementary Note 2)." concerning the Minor point.

Therefore, we added the Supplementary Note 2 "Simplified scheme with an auxiliary magnetic field instead of magnetic anisotropy": "We note that the case of Co adatom on Cu(111) was addressed in the preliminary work reported in Ref. [7], where the self-energies of the adatoms were computed in a very simplified approach compared to the current work. The results are, however, in line with those obtained with the more accurate formalism described in our manuscript. In Ref. [7], a rudimentary projection basis (based on regular scattering solutions that are computed at the Fermi energy) was utilized, which is not that accurate to describe various spin-dynamics properties. However, and more importantly it was lacking the impact of the spin-orbit coupling. The simplest implication of the latter is to open a gap of a few meV in the susceptibility, which has to be precisely described. Instead of spin-orbit coupling, a magnetic field was utilized to effectively open a gap in the spin-excitations spectra. The former has profoundly different implications on the dynamical susceptibilities compared to those induced by the latter. The main ingredient is the definition of the correct ground state of the magnetic orientation with respect to the lattice, which is not known when magnetic fields are used. Additionally, the magnitude of the field is arbitrary and has a different physical origin. These aspects induce drastic changes in the properties of the susceptibilities — and therefore in the self-energy and the resulting renormalized electronic structure: the shape and weight of the spin-flip relativistic responses, χ^{+-} and χ^{-+} , at positive and negative frequencies (both imaginary and real parts) change dramatically if the moment is lying in an easy-plane or along an easy axis, depending also if it is in the ground state or in a metastable state. Consequently, it is impossible to make any reasonable claim and systematic interpretation of the experimental data of Co on the three noble metallic substrates (Cu, Ag, Au) from such a poor's man approach. These issues were solved by our current approach. "

while for the Major point we added in the Discussion section: "While the self-energies quantifying the interaction of the electrons and spin-excitations are dynamical in nature and account for various correlation effects, it would be interesting to prospect in the future the impact of correlations (in the spirit of DFT + U [Ref.] on the ground state properties, such as the magnetic anisotropy energy and subsequently on the excitation behavior of the investigated materials."

Third Reviewer

Due to the revisions and replies made by the authors I think I understand better at least than before, for which I appreciate the authors' efforts. The authors demonstrate in this manuscript that the tunneling spectra taken on Co adsorbates on noble metal (111) surfaces, which have been believed Kondo resonance in surface science community, can be explained with a gap due to spin-flipping inelastic tunneling plus a bound state called spanaron. The scenario the authors propose could be true, but it seems to me dangerous at the present stage to publish it in the prestigious journal and to provide a kind of scientific approval to the community. One of the major reasons of the my hesitate about the acceptance is lack of experimental supports. The authors nicely fit the spectra with the their model, but the spectra were also nicely explained

Prof. Dr. Samir Lounis

Chair of theoretical nanospintronics at RWTH-Aachen
Leader of Functional Nanoscale Structure Probe and
Simulation Laboratory
Tel.: 02461-61-4068
E-Mail: s.lounis@fz-juelich.de
www.fz-juelich.de/pgi/Group-Lounis

PGI-1/IAS-1: Quantum Theory of Materials

with the Kondo model. According to the authors' calculation results, Co adatoms exhibit out-of-plane magnetization, which is obviously incompatible with the Kondo model. The presence of the magnetization can be detected by using spin-polarized STM presumably without difficulty. XMCD may also can be used for the detection. Whereas the spectral feature is "a dip" for the Kondo model, it is "a gap" for the authors model. The temperature broadening thus should behave quite different from each other. Taking temperature-dependent tunneling spectra, which is not very difficult using advanced commercial STM setups, will provide a clear answer which is plausible. It seems the authors already contacted with some leading and pioneering researchers in this field. I wonder why they do not provide experimental results supporting the scenario. If experimental results inconsistent with the Kondo model are presented it will be immediately accepted without hesitation.

Answer 3.1: We appreciate that the Reviewer agrees that "we demonstrate that the tunneling spectra of Co on adatoms on noble metals, which have been believed to be Kondo resonances in the surface science community, can be explained with a gap due to spin-flipping inelastic tunneling plus a bound state called spinaron."

First, we would like to correct your statements that "we nicely fit the spectra with our model." We do not proceed to a fit but use a state-of-the-art ab-initio scheme to get theoretical STM spectra, which strikingly match the experimental spectra. There are no adjustable parameters involved. Even more impressive is that this is not only working for a single case but on three different surfaces: Cu(111), Ag(111) and Au(111). Following the suggestion of the second Reviewer, our ab-initio simulations provide a reasonable description of the observed features on Co/Cu(001), Co/Ag(001) and even Ti/Ag(001). This is not a coincidence and it is, so far, the best demonstration that our ab-initio simulations are reliable in explaining the observed features. We highlight that we are not aware of any ab-initio scheme capable to provide such a one-to-one correspondence.

In our manuscript, we already make several experimental proposals on how to check our predictions (more precisely, on the subsection "Impact of spin-polarized tip, magnetic field and proximity-effects"). Following the comment of the Reviewer, we modified the title of subsection to "Experimental proposals: Impact of spin-polarized tip, magnetic field and proximity-effects" in order to better convey its goal to the reader. Before addressing those proposed experiments, it is crucial to understand what were/are the current STM limitations in proving that the observed zero-bias anomalies (ZBAs) are Kondo resonances. Fitting the observed features with Fano-shapes is not enough to prove that these are Kondo resonances. We did a similar fit on our theoretical spectra and got very similar results to those reported experimentally (see Table 2 in the main text).

The Reviewer is right to propose to use temperature to track the behavior of the broadening of the features. The question is: why this has never been reported say for Co/Cu(111) or for Co/Au(111), while the first experimental measurements were done in 1998? We learned from our experimental colleagues that the reason is that the ZBAs are rather broad and current experimental resolution do not permit to track their changes upon increase of temperature. From our theory, the broadening is linear with the MAE and finds its origin in the density of electron-hole excitations. Resolution aspects can be improved in the future of course, maybe by using electron-spin-resonance STM [Balatsky, A. V. *et al.* *Advances in Physics* 61, 117–152 (2012);

Prof. Dr. Samir Lounis

Chair of theoretical nanospintronics at RWTH-Aachen
Leader of Functional Nanoscale Structure Probe and
Simulation Laboratory
Tel.: 02461-61-4068
E-Mail: s.lounis@fz-juelich.de
www.fz-juelich.de/pgii/Group-Lounis

PGI-1/IAS-1: Quantum Theory of Materials

Baumann et al. *Science* 350, 417 (2015)], which is however not yet achieved on adatoms sitting directly on a metal. Also spin-polarized STM cannot be easily used to get a spin-polarized spectrum of a single adatom, e.g. using the so-called STM-based magnetometry (see for example Ref. [Meier et al. *Science* 320, 82 (2008)], or at least not the way this can be done on a magnetic monolayer. However, we made proposals in our manuscript on how this can be achieved. For instance, since our theoretical spectra are different for both spin channels, we expect their spin-dependent weight to the total spectrum to change depending on the spin-polarization of the STM tip. Another experiment based on our predictions: if one reduces the MAE of the Co adatom, which might happen by attaching it to a non-magnetic atom such as Cu (or some other atom), the gap of the spin-excitations could be reduced, which would then enhance the chances of increasing their lifetimes and thus improve the ability to track the temperature dependence.

Besides the temperature dependence, we also proposed to use a magnetic field, which is expected to split a Kondo feature, while it should increase the gap of the regular spin-excitations. In the particular cases addressed in our manuscript, the presence of the spinaron leads to an unusual behavior: the gap increases but the minimum of the gap shifts to positive bias voltages upon application of a magnetic field. This is a striking behavior, which should be detected by STM setups with large magnetic fields. We cited in the main manuscript papers reporting on such setups. We also proposed to use proximity effects to induce large effective magnetic fields, which should lead to the predicted behavior. For instance, a magnetic atom, known to not show spin-excitations and no Kondo behavior [Khajetoorians *et al.* *Nature Physics* volume 8, 497 (2012)] can be brought at the vicinity of the Co adatom, without being nearest neighbors such that its electronic structure is not altered.

Furthermore, we agree with the Reviewer that XMCD would be a great tool to prospect the magnetic of Co adatoms on noble metals, which surprisingly was not undertaken. At this stage, we can only conjecture that the community avoided to look at those systems since they were accepted to be conventional Kondo cases, and thus of no interest for XMCD measurements.

We believe that our proposals will guide current and future experimental work and certainly our work motivated a new interest in the real understanding of the ZBAs.

Here are the amendments made to the manuscript:

- At the end of the Introduction "**Finally, we propose possible experiments that enable the verification of the origin of the investigated zero-bias anomalies.**"
- At the beginning of the subsection "Experimental proposals: Impact of spin-polarized tip, magnetic field and proximity-effects", we added: "**As mentioned before, the Kondo origin of the low-energy spectral features has so far not been evidenced for the systems investigated in the current work. This is usually realized by performing temperature-dependent measurements and/or upon the application of a magnetic field, which would respectively result in a broadening of the anomalies and/or their splitting. However, the large broadening of the dip-like structure require improved energy resolutions than currently available, preventing the realization of such experimental studies. Here we address possible experiments that can further verify our predictions.**"
- At the end of the previous subsection: "**If the MAE is reduced, the lifetime of the spin-excitations is expected to increase, since the amount of electron-hole excitations available in the respective energy range would decrease. This can then favor the monitoring of the impact of temperature and magnetic**

Prof. Dr. Samir Lounis

Chair of theoretical nanospintronics at RWTH-Aachen
Leader of Functional Nanoscale Structure Probe and
Simulation Laboratory
Tel.: 02461-61-4068
E-Mail: s.lounis@fz-juelich.de
www.fz-juelich.de/pgi/Group-Lounis

PGI-1/IAS-1: Quantum Theory of Materials

field on the zero-bias anomalies, helping to distinguish a Kondo behavior from the one emerging from spin-excitations."

- In the Discussion section "The one-to-one agreement between our first-principles spectra and the available experimental ones strongly advocates for the importance of the spin-excitations in the interpretation of the origin of the zero-bias anomalies. X-ray magnetic circular dichroism (XMCD) experiments should help to confort our findings by unveiling the magnetic nature as well as the magnetic anisotropy energy of the investigated adatoms as done for Co adatoms on Pt(111) [Ref.]. Surprisingly, this was, so far, not performed. Temperature-dependent and magnetic-field STM-based measurements were, to our knowledge, not reported, which is explained by the extreme difficulty to probe with enough resolution modifications induced in the rather broad spectral features. We conjecture that this might change in the near future, for example with electron-spin-resonance STM (ESR-STM) [Refs.] if realized on metallic substrates. In this work, various experimental setups were proposed, which would permit to further confirm our predictions. For instance, the theoretical spectra are spin-dependent and therefore the weight of each spin-channel to the total STM spectrum should depend on the spin-polarization of the tip. Furthermore, the application of a magnetic field is expected to increase the gap of the intrinsic spin-excitations, while a splitting is expected for Kondo features. However, the presence of the spinaron leads to an unconventional behavior, that is the excitation gap increases but the effective dip is not fixed and shifts to larger bias voltages. Currently, a few STM setup allow to reach large magnetic fields (e.g., 14 T and even 38 T), which would be enough to check our predictions. But even if those fields are not available, a reasonable alternative would be to use the proximity-induced effective magnetic field emerging from an adjacent magnetic adatom. Finally, one could tune down the magnetic anisotropy energy in order to reduce the amount of electron-hole excitations that are responsible for the broadening of the spin-excitations. This could be realized by attaching a non-magnetic atom such as Cu, for example, to Co adatom, after which the experimental investigation of the impact of temperature and magnetic fields would become more amenable."